# Communication between DNA polymerases and Replication Protein A within the archaeal replisome

Markel Martínez-Carranza [1,9], Léa Vialle[2,9], Clément Madru [1,9], Florence Cordier [3,4,9], Ayten Dizkirici Tekpinar [1,8], Ahmed Haouz [5], Pierre Legrand [6], Rémy A. Le Meur[3], Patrick England [7], Rémi Dulermo[2], J. Iñaki Guijarro [3], Ghislaine Henneke[2] ✉ & Ludovic Sauguet [1] ✉

Replication Protein A (RPA) plays a pivotal role in DNA replication by coating and protecting exposed single-stranded DNA, and acting as a molecular hub that recruits additional replication factors. We demonstrate that archaeal RPA hosts a winged-helix domain (WH) that interacts with two key actors of the replisome: the DNA primase (PriSL) and the replicative DNA polymerase (PolD). Using an integrative structural biology approach, combining nuclear magnetic resonance, X-ray crystallography and cryo-electron microscopy, we unveil how RPA interacts with PriSL and PolD through two distinct surfaces of the WH domain: an evolutionarily conserved interface and a novel binding site. Finally, RPA is shown to stimulate the activity of PriSL in a WH-dependent manner. This study provides a molecular understanding of the WH-mediated regulatory activity in central replication factors such as RPA, which regulate genome maintenance in Archaea and Eukaryotes.

Single-stranded DNA-binding proteins (SSBs) are essential components of the DNA replication machinery that coat and protect exposed single-stranded DNA (ssDNA) in all domains of life[1,2]. In Bacteria, the archetypal SSB is the major single-stranded DNA-binding protein. It contains a single oligonucleotide/oligosaccharide-binding domain (OB domain) and assembles into homotetrameric complexes. Eukaryotes also encode single OB-fold SSBs, but their function is restricted to DNA damage repair, whereas the main ssDNA-binding component of the replisome is the heterotrimeric RPA complex (Replication Protein A). In recent years, RPA has also been shown to be a central regulator of eukaryotic DNA metabolism, acting as a molecular hub that coordinates the recruitment and exchange of genomic maintenance factors[3]. In eukaryotes, RPA participates in both the initiation and elongation

steps of DNA replication by enhancing the assembly and recruitment of DNA polymerases, promoting polymerase switch on the lagging strand and by coordinating the processing of Okazaki fragments[4–7]. Furthermore, DNA replication and genotoxic stresses are signaled throughout the cell cycle via the detection of persistent stretches of RPA-ssDNA[8]. RPA is also implicated in several DNA repair pathways, during nucleotide excision repair, base excision repair, mismatch repair and homologous recombination[9,10].

While archaeal chromosomes resemble those of most Bacteria, their DNA replication machinery is more closely related to their eukaryotic counterparts, serving as powerful models for understanding the function and evolution of the eukaryotic replication machinery. We have previously shown that RPA from *Pyrococcus abyssi*

[1]Architecture and Dynamics of Biological Macromolecules, Institut Pasteur, Université Paris Cité, CNRS UMR 3528, Paris, France. [2]Univ Brest, Ifremer, CNRS, Biologie et Ecologie des Ecosystèmes marins profonds (BEEP), Plouzané, France. [3]Biological NMR & HDX-MS Technological Platform, Institut Pasteur, Université Paris Cité, CNRS UMR 3528, Paris, France. [4]Structural Bioinformatics, Institut Pasteur, Université Paris Cité, CNRS UMR 3528, Paris, France. [5]Crystallography Platform, C2RT, Institut Pasteur, Université Paris Cité, CNRS UMR 3528, Paris, France. [6]Synchrotron SOLEIL, HelioBio group, L'Orme des Merisiers, Saint-Aubin, France. [7]Molecular Biophysics Platform, C2RT, Institut Pasteur, Université Paris Cité, CNRS UMR 3528, Paris, France. [8]Present address: Department of Molecular Biology and Genetics, Van Yüzüncü Yil University, Van, Turkey. [9]These authors contributed equally: Markel Martínez-Carranza, Léa Vialle, Clément Madru, Florence Cordier. ✉e-mail: ghislaine.henneke@ifremer.fr; ludovic.sauguet@pasteur.fr

forms a heterotrimer displaying close homology to eukaryotic RPA. The core of the archaeal replisome[11] hosts a CMG-like 3'-5' replicative helicase (Cdc45-MCM-GINS) that shares a similar architecture with eukaryotic CMG (Fig. 1a), yet distinguishing itself in two main features: archaeal MCM forms a homo-hexamer, and most archaeal replicative helicases contain a nuclease named GAN (GINS-associated nuclease), which is orthologous to eukaryotic Cdc45 and bacterial RecJ[11]. Most Archaea also encode eukaryotic-like heterodimeric DNA primases, PCNA sliding-clamps, and RFC clamp loaders[11,12] (Fig. 1b). Additionally, while all Archaea encode at least one copy of a canonical replicative polymerase PolB, most Archaea use an archaeal-specific DNA polymerase[13] named PolD[14–17].

Several winged-helix domains (WH) are found among these replication factors (Fig. 1b). In the Orc1-Cdc6 DNA replication initiation factor, its WH domain contributes to origin DNA binding[18]. The C-terminal WH domain of MCM has been shown to interact with Orc1[19] and to be an allosteric regulator of both the ATPase and helicase activities of MCM from *Saccharolobus solfataricus*[20]. Former bioinformatic studies have shown that the C-terminal region of Rpa2 also hosts a WH domain, but its structure has not yet been reported[21]. Crystal and cryo-EM structures of the RPA heterotrimeric complex of *P. abyssi* were recently determined by our group in the presence and absence of DNA[22]. The C-terminus of Rpa2 (residues 186–268) could not be resolved in any of these structures (Fig. 2a). In eukaryotes, the WH domain of Rpa2 (Rpa2WH) has been shown to recruit several key enzymes involved in DNA damage response[23–26], and to contribute to protein-protein interactions that are essential for primosome assembly during SV40 viral DNA replication[27,28]. In agreement with this central role in eukaryotic DNA metabolism, yeast truncation mutants lacking the C-terminal domain of Rfa2 (yeast ortholog of Rpa2) are hypersensitive to DNA damaging agents and exhibit mutator and hyper-recombination phenotypes[29,30].

The current study sheds light on the biological role of the evolutionarily conserved winged-helix domain[31] of RPA, in recruiting and modulating the activities of the two main archaeal replicative DNA polymerases: the DNA primase (PriSL) and PolD. The functional implications of these interactions were further investigated in vivo through genetic studies in *Thermococcus barophilus* and in vitro using polymerization activity assays, as well as protein-protein interaction experiments. DNA priming and primer-extension activity assays revealed that PriSL is stimulated by RPA through a WH-dependent mechanism that we termed the 'WH-bait' model. In addition, by using an integrative structural biology approach combining Nuclear Magnetic Resonance (NMR), X-ray crystallography and cryo-electron microscopy, we uncovered the structural basis for the interaction between RPA and these two primordial replicative polymerases. We infer that WH domains present in RPA, which are conserved in archaea and eukaryotes, play a pivotal role in polymerase recruitment and switching during DNA replication and repair.

## Results

### The C-terminal domain of *P. abyssi* Rpa2 contains a conserved WH domain

We determined the solution structure of the C-terminal region of Rpa2 comprising residues 178–268 by NMR (Fig. 2b, c, Supplementary Fig. 1 and Supplementary Table 1). The backbone dynamics (order parameter $S^2$ and exchange contribution $R_{ex}$) were further investigated by $^{15}N$ relaxation measurements (Supplementary Figs. 2 and 3). The structural ensemble of the Rpa2 C-terminal region shows a convergent structured domain (206-268) composed of a three-helix bundle and a short three-stranded antiparallel β sheet (Fig. 2b). This domain belongs to the WH-like DNA-binding domain superfamily[31] (Rpa2WH). Rpa2WH behaves as a rigid globular domain, with $S^2$ values around $0.88 \pm 0.06$ in the three α-helices and $0.85\pm0.05$ in the three β-strands, and contains a flexible hotspot encompassing the $\alpha_1/\beta_1$ and the β-wing loops ($S^2 < 0.75$) (Supplementary Fig. 3). In contrast, the N-terminal region (178–205) is essentially disordered as indicated by the lack of distance constraints that results in poor convergence of the structures (high backbone root mean square deviations) (colored in red in Fig. 2c). The N-terminal disorder is caused by very high-amplitude motions on the ps-ns timescale as evidenced by the low $S^2$ values gradually decaying towards zero before residue E206. This observation is consistent with the proposal that this region may be a flexible linker to the OB-3 ssDNA-binding domain (residues 40–171) of Rpa2[22]. Having solved the structure of the isolated Rpa2WH, we set out to study its role in vivo.

### Truncating the Rpa2WH domain is possibly lethal in *Thermococcus barophilus*

Recently, effective genetic tools have been developed in *T. barophilus* to study genomic maintenance under extreme temperature and/or high hydrostatic pressure conditions found in deep-sea hydrothermal vents[32]. The *rpa1*, *rpa2* and *rpa3* genes, which encode proteins that share ~70% sequence identity with *P. abyssi* RPA, were targeted with primers designed to delete the genes (Fig. 2d). As expected, multiple RPA deletion attempts were unsuccessful, suggesting that this primordial replication factor is essential for *T. barophilus* cell growth (5 transformations, 17 clones screened). The region of the *rpa2* gene that encodes the C-terminal WH domain of RPA was also targeted for deletion.

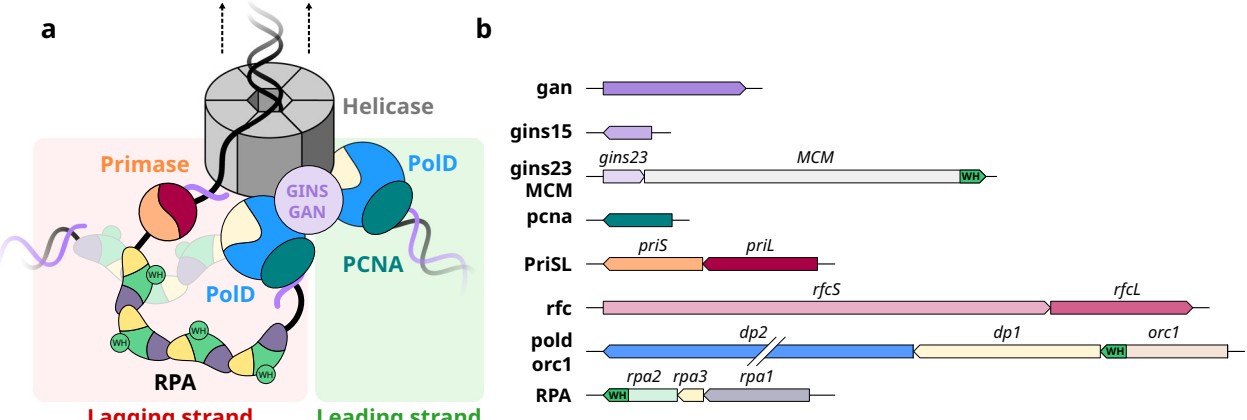

**Fig. 1 | Winged-helix domains in the archaeal replisome. a** Schematic representation of the archaeal core replisome and RPA binding single-stranded DNA on the lagging strand. **b** Genes from *Pyrococcus abyssi str. GE5* that encode DNA replication factors composing the replisome. Genes encoding a protein that contains a winged-helix domain are highlighted in green.

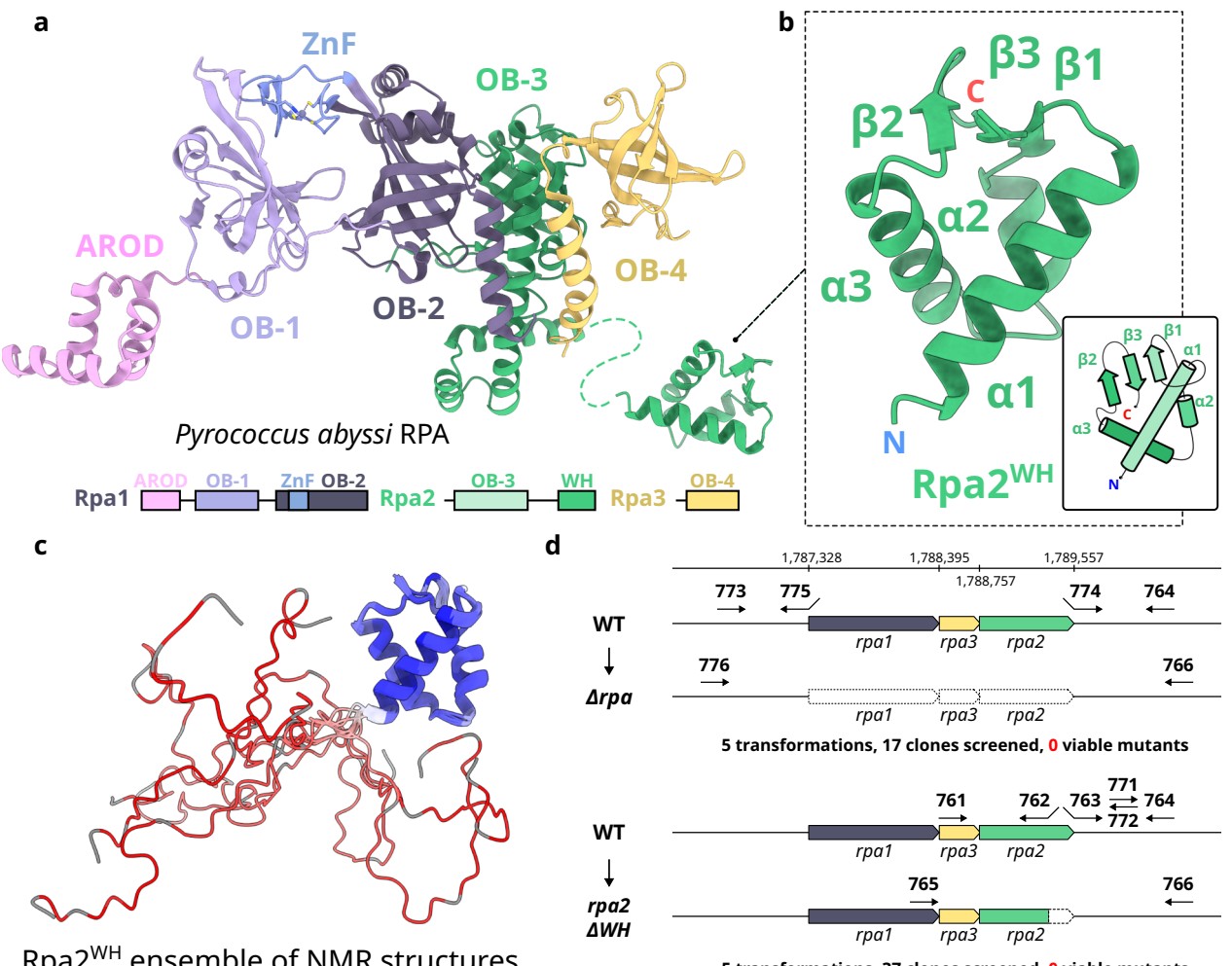

**Fig. 2 | The C-terminal region of Rpa2 contains a conserved WH domain.**
**a** Cartoon representation of *P. abyssi* RPA (PDB ID 8AAJ[22]). **b** Focus on Rpa2[WH], for which a representative structure from the NMR ensemble is shown. **c** NMR structural ensemble of Rpa2[WH], color coded with $S^2$ values from red to blue denoting high and restricted amplitude motions respectively (gray: no value). **d** Schematic representation of the *rpa* locus. All the primers used to construct mutants are indicated, as well as the number of transformation assays and screened clones for both expected mutants: deletion of the three *rpa* genes (top) and the WH domain of Rpa2 (bottom).

Interestingly, all attempts to generate WH truncated RPA variants were also unviable (5 transformations, 37 clones screened), suggesting that Rpa2[WH] plays an essential role in *Thermococcales* Archaea. A genetic study on the methanogenic archaeon *Methanococcus maripaludis* found that *rpa1* and *rpa2* are possibly essential genes, whereas *rpa3* is not, highlighting the critical biological role of Rpa2[33].

A recent study conducted by our group[22] showed that archaeal Rpa2[WH] is not primarily involved in DNA binding but rather in protein-protein interactions. Former studies on *P. abyssi* and *T. kodakarensis* have shown that RPA interacts with DNA polymerases[34,35], suggesting that it may participate in the normal progression of the replication fork. Given its potential essentiality, we hypothesized that the Rpa2[WH] domain could be responsible for the communication between RPA and the replicative DNA polymerases.

## RPA interacts with PriSL and PolD through Rpa2[WH]

Our group recently described the distinct oligomeric states adopted by *P. abyssi* RPA in the presence and absence of ssDNA[22]. In the absence of ssDNA, RPA forms a tetrameric supercomplex that occludes its DNA-binding sites, disassembling into nucleoprotein filaments in the presence of ssDNA[22]. In bio-layer interferometry (BLI) protein-protein interaction assays, we confirmed that deletion of Rpa2[WH] completely disrupts the interaction of immobilized RPA with two key archaeal

replicative DNA polymerases: PriSL and PolD (Fig. 3a, c). Additionally, RPA-bound nucleoprotein filaments lacking Rpa2[WH] displayed seven-fold and fourfold weaker interactions with PriSL and PolD respectively under the tested conditions (12.5 nM PriSL in Fig. 3b, 250 nM PolD in Fig. 3d). In order to measure the affinity of these interactions, steady-state kinetic analyses of immobilized Rpa2[WH] and immobilized RPA-ssDNA nucleofilaments were performed with PriSL (Fig. 3e, f) and PolD (Fig. 3g, h). Our results show that Rpa2[WH] and RPA-ssDNA nucleofilaments display similar affinity for PriSL ($K_D = 25 \pm 9$ nM and $24 \pm 2$ nM, respectively) and for PolD ($K_D = 98 \pm 49$ nM and $136 \pm 17$ nM, respectively), indicating that Rpa2[WH] is the main contributor to these protein-protein interactions.

The interfaces between Rpa2[WH] and the two DNA polymerases were further delineated in solution using NMR spectroscopy (Fig. 3i, j), by monitoring ¹H and ¹⁵N Chemical Shift Perturbations (CSP) and peak intensity changes in the ¹⁵N-Heteronuclear Single-Quantum Coherence (HSQC) spectra of Rpa2[WH] upon addition of unlabeled PriSL[ΔCTD] or PolD (Supplementary Fig. 4). Addition of a 0.5 molar ratio of PriSL[ΔCTD] or 0.1 molar ratio of PolD is already sufficient to induce a drastic decrease in the peak intensities of many residues as well as small CSP. Overall, the peak intensity ratio between PriSL[ΔCTD]-bound and free Rpa2[WH] are below 0.4 in the region 200–268, suggesting that the globular domain of Rpa2[WH] and the neighboring linker residues 200-205 behave like a high

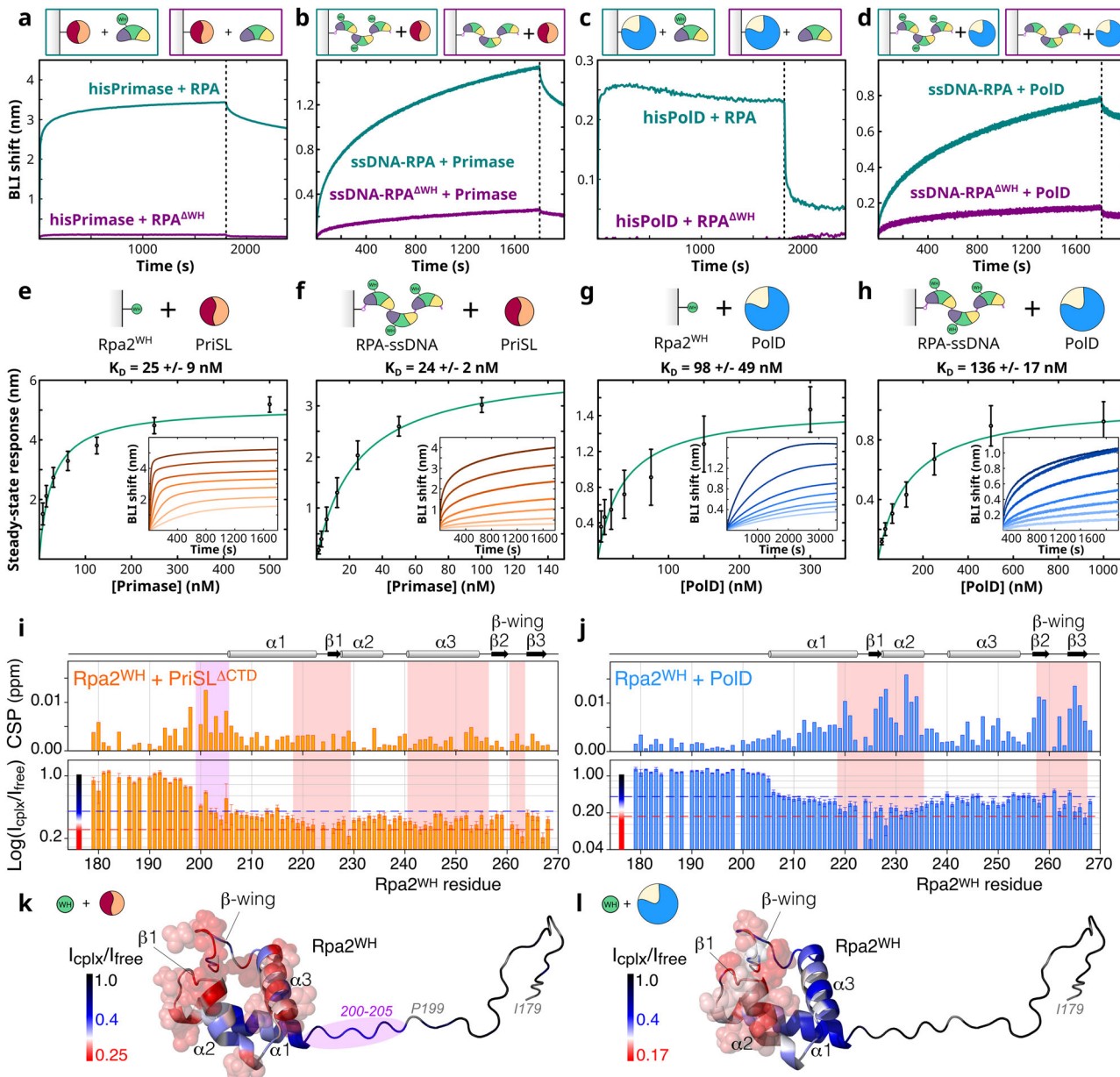

**Fig. 3 | Rpa2^WH connects RPA to the archaeal replicative DNA polymerases.**
**a**–**h** Biolayer Interferometry (BLI) results for (**a**) specific binding of immobilized histidine-tagged PriSL at 50 nM to 10 μM wild-type RPA (teal) and to 10 μM RPA^ΔWH (purple); **b** specific binding of immobilized biotin-tagged nucleoprotein RPA filaments to 12.5 nM PriSL (teal) and immobilized biotin-tagged nucleoprotein RPA^ΔWH filaments to 12.5 nM PriSL (purple); **c** specific binding of immobilized histidine-tagged PolD at 50 nM to 1 μM wild-type RPA (teal) and to 1 μM RPA^ΔWH (purple); **d** specific binding of immobilized biotin-tagged nucleoprotein RPA filaments to 250 nM PolD (teal) and immobilized biotin-tagged nucleoprotein RPA^ΔWH filaments to 250 nM PolD (purple). **e** Specific binding of PriSL (500, 250, 125, 62.5, 31.25, 15.62, 7.81 nM, *n* = 3) to immobilized histidine-tagged Rpa2 C-terminal winged-domain. **f** Specific binding of PriSL (100, 50, 25, 12.5, 6.25, 3.12, 1.56 nM, *n* = 3 biological replicates) to immobilized biotin-tagged nucleoprotein RPA filaments. **g** Specific binding of PolD (1000, 500, 250, 125, 62.5, 31.25, 15.62 nM, *n* = 3 biological replicates) to immobilized histidine-tagged Rpa2^WH. **h** Specific binding of PolD (1000, 500, 250, 125, 62.5, 31.25, 15.62 nM, *n* = 3) to immobilized biotin-tagged nucleoprotein RPA filaments. Steady-state analyses were performed using the average signal measured at the end of the association steps. Data are represented as mean value ± standard deviations (error bars). Raw data are provided in the source data file. **i**–**l** Identification of the Rpa2^WH binding surface to the DNA polymerases by NMR. **i**, **j** NMR chemical shift perturbations (CSP) and peak intensity ratios $I_{cplx}/I_{free}$ (log₁₀ scale) on Rpa2^WH induced by PriSL^ΔCTD and PolD, respectively. The dotted blue and red lines correspond to $I_{cplx}/I_{free}$ ratios of 0.4 and 0.25 for the complex with PriSL^ΔCTD and 0.4 and 0.17 for the complex with PolD, as used for the color coding in (**k**, **l**). Error bars in the $I_{cplx}/I_{free}$ histograms represent the noise standard deviation in the spectra (see section "Methods"). For clarity, the most affected regions in terms of CSP and/or intensity ratio are highlighted by light red boxes. Secondary structure elements are indicated at the top. **k**, **l** Mapping of the intensity ratio $I_{cplx}/I_{free}$ on Rpa2^WH color coded from black (no attenuation), blue (weak attenuation) to red (large attenuation) as indicated. Residues in gray denote missing data (proline or overlapping signals). The unfolded residues 200-205 that transiently contact PriSL^ΔCTD are highlighted in light violet. The residues that are most severely affected by the binding ($I_{cplx}/I_{free}$ < 0.29 for PriSL^ΔCTD and <0.26 for PolD) are depicted as transparent spheres and delineate the respective binding surfaces to the polymerases.

molecular weight complex, whereas the remaining unfolded residues (179–199) are essentially unaffected by the binding (intensity ratio -1). In the experiment with PolD, only the globular domain of Rpa2$^{WH}$ (206–268) is affected by the binding (intensity ratio <0.4), the unfolded region 179-205 being totally unaffected (intensity ratio -1). This indicates that the interaction of Rpa2$^{WH}$ with its partners PriSL$^{\Delta CTD}$ and PolD is primarily mediated by the globular region of the Rpa2$^{WH}$ domain. Mapping the small variation of intensity ratio (in the range 0.25–0.4 for PriSL$^{\Delta CTD}$ and 0.17–0.4 for PolD, Fig. 3k, l) within the folded domain of Rpa2$^{WH}$ delineates two partially overlapping but distinct binding surfaces with the DNA polymerases. Interestingly, no substantial binding was observed between Rpa2$^{WH}$ and PolB (Supplementary Fig. 5), the third replicative DNA polymerase in *P. abyssi*. No significant and localized CSP or peak intensity change could be detected upon addition of equimolar amounts of PolB in $^1$H-$^{15}$N HSQC spectra.

In conclusion, Rpa2$^{WH}$ is responsible for the binding of RPA to PolD and PriSL, both in the absence and in the presence of ssDNA. However, the molecular interface of Rpa2$^{WH}$ with these polymerases and the effect of RPA on their biological activity remain unknown.

### RPA stimulates the progression of PriSL through a WH-dependent mechanism

The impact of RPA binding on PolD and PriSL activities was examined through primer extension activity assays using denaturing gel electrophoresis (Fig. 4), where a fluorescently labeled 17 nucleotide-long DNA primer annealed to a 87 nucleotide-long template was extended by PolD or PriSL for 10 min at 55 °C in presence of ribonucleotides and deoxyribonucleotides at physiologically relevant concentrations (see materials and methods section). In the absence of RPA, PolD readily extends the fluorescently labeled primer, resulting in the synthesis of 87-nt extension products (Fig. 4a). The addition of RPA leads to a reduction in the amount of 87-nt extension products compared to the reaction without RPA. The primer extension activity of PolD is progressively hindered by RPA in a concentration-dependent manner, leading to a loss of full-length primer extension. Obstruction of PolD also leads to an accumulation of shorter digested products, consistent with increased exonuclease activity when primer elongation is hindered, such as during dNTP depletion[17]. In reactions containing only Rpa2$^{WH}$, no major change was observed in the length of elongation products (Fig. 4b). However, the negative effect was restored in reactions with truncated RPA$^{\Delta WH}$ (Fig. 4c) as well as in reactions containing both RPA$^{\Delta WH}$ + Rpa2$^{WH}$, resulting in nearly similar reduced amounts of full-length 87-nt extension products (bar graphs in Fig. 4c, d). The observed effect of RPA on elongated product size by PolD suggests that RPA-bound ssDNA obstructs PolD's elongation activity in a way that is independent of Rpa2$^{WH}$. However, it is important to note that this inhibitory effect observed under our experimental conditions may be mitigated in a cellular context by the presence of other replication factors, such as PCNA[36].

On the other hand, the elongation activity of PriSL was notably enhanced up to fivefold with increasing amounts of RPA (Fig. 4e), in line with our past results[36]. In this case, PriSL was capable of primer extension with the accumulation of longer products (approximately ≥70 nt in length). Two additional pause sites at -65-nt and -64-nt were also observed, indicating aborted DNA synthesis. It is worth noting that although PolD was able to elongate the primer up to the length of the template (87 nt), PriSL alone showed a distinct elongation product profile consisting of two bands between 57 and 87 nucleotides. Addition of full-length RPA resulted in the appearance of a new longer product, that was enriched in an RPA concentration-dependent manner. The addition of isolated Rpa2$^{WH}$ did not affect PriSL activity, suggesting that the RPA DNA-binding core is required to stimulate the elongation activity of PriSL (Fig. 4f). Interestingly, the deletion of the Rpa2$^{WH}$ domain (RPA$^{\Delta WH}$) led to a drastic loss of long elongation products (≥70 nt in length) by PriSL, indicating that

in absence of its Rpa2$^{WH}$ domain, RPA hinders the progression of PriSL (Fig. 4g), similarly to what we observed for PolD. The length of the products rapidly decreased with increasing amounts of RPA$^{\Delta WH}$, resulting in the accumulation of shorter products (-20 nt in length) and the loss of the longest products (-64-nt and 65-nt pausing sites and ≥70 nt in length). The stimulatory effect of RPA on PriSL extension activity could not be restored by adding Rpa2$^{WH}$ together with RPA$^{\Delta WH}$ (Fig. 4h).

In addition, DNA priming activity assays were conducted on a pM13 circular ssDNA substrate, with increasing concentrations of RPA and physiological concentrations of nucleotides. Importantly, the priming activity assay allows for the visualization of both priming and elongation. PriSL readily initiates and extends fragments up to 200-nt in length in the absence of RPA (Supplementary Fig. 6a). Addition of RPA was responsible for the synthesis of longer products (200 to 1000 nt in length) by PriSL, in agreement with the primer extension experiments. Our results indicate that RPA affects PriSL activity favoring primer elongation rather than priming frequency in our reaction conditions.

### PriSL preferentially incorporates dNTPs during primer elongation

While the eukaryotic PriSL synthesizes a short RNA primer, the archaeal primase has been shown to synthesize a 5′-RNA/DNA-3′ mixed primer by stochastically incorporating dNTPs during priming but exclusively using dNTPs during elongation[37]. To account for these dual synthetic modes, our in vitro activity assays used a mix of dNTPs and rNTPs, with concentrations experimentally determined to reflect the nucleotide levels in *P. abyssi* in vivo[38]. To assess whether RPA stimulation affects PriSL selectivity and the nucleotide composition of the newly synthesized strand, all primer-extension experiments were repeated with both full-length and truncated RPA constructs. For each reaction, half of the mixture was incubated at 55 °C for 2 h with either 250 mM NaCl or 250 mM NaOH (Supplementary Fig. 7). In the positive control, alkaline treatment completely degraded a DNA substrate containing a single ribonucleotide (Supplementary Fig. 7i). However, when the reaction mixtures were analyzed on a denaturing gel, no bands were lost or degraded after alkaline treatment (Supplementary Fig. 7a–h). Altogether, these experiments confirm that *P. abyssi* PriSL extends a primer using dNTPs in the presence or absence of RPA. The priming activity assays were also repeated exclusively with either rNTPs or dNTPs (Supplementary Fig. 6b, c). As expected, these results show that dNTPs are required for optimal primer extension. Nevertheless, these experiments demonstrate that RPA stimulates primer extension by the primase across all nucleotide combinations: dNTPs, rNTPs, or dNTPs+rNTPs.

Altogether, both priming and extension activity assays indicate that RPA enhances the synthesis of longer products by PriSL in a WH-dependent manner. This stimulatory effect contrasts with the WH-independent reduced activity reported for PolD. To gain insights into the molecular basis underlying these effects, the structures of PriSL and PolD were determined in complex with the Rpa2$^{WH}$ domain.

### Rpa2$^{WH}$ binds to the catalytic PriS subunit of the DNA primase

The crystal structures of PriSL$^{\Delta CTD}$ in its apo form and bound to Rpa2$^{WH}$ were determined at 1.85 Å and 3.5 Å resolution, respectively (Supplementary Table 2). For crystallization purposes, the C-terminal domain of the PriL subunit was deleted, which contains a 4Fe-4S cluster with a debated biological function[39]. Importantly, Rpa2$^{WH}$ was shown to bind to both full-length PriSL (Fig. 3e) or PriSL$^{\Delta CTD}$ with a similar nanomolar affinity[22], indicating that the PriL$^{CTD}$ domain does not contribute to the interaction with RPA. The reconstruction of the complex was facilitated by docking the high-resolution crystallographic structure of PriSL$^{\Delta CTD}$ and the NMR structure of the Rpa2$^{WH}$ domain into the electron density map (Fig. 5a and Supplementary Fig. 8).

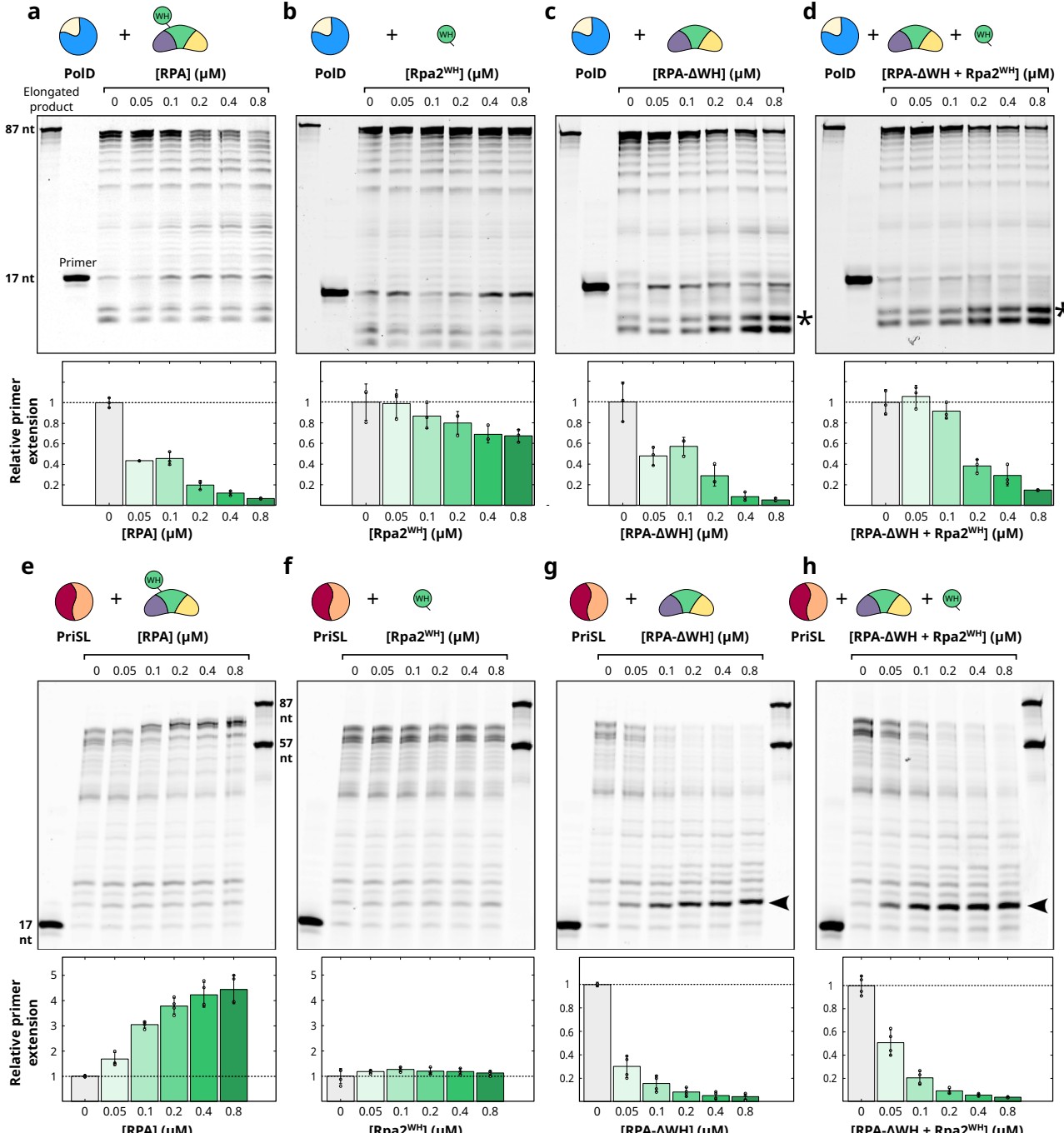

**Fig. 4 | Impact of RPA binding on PolD and PriSL primer extension activity.** A 17 nucleotide-long primer labeled with a 5′ Cy5 fluorophore annealed to a 87 nucleotide-long template was used as substrate in reactions with all deoxyribonucleotides and ribonucleotides at physiological concentrations (sequences are shown in Supplementary Table 4). Reactions were incubated for 10 min at 55 °C with PolD or PriSL, and one of several RPA constructs. Experiments were repeated $n = 4$ times, band integration was performed in all 4 biological replicates (87 nt band for PolD gels, >70 nt bands for PriSL gels) to derive standard deviation. Each bar shows the mean value, standard deviation is represented as error bars, and individual measurements are shown as white dots. Uncropped gels are provided as a Source Data file. **a**–**d** Impact of RPA binding on PolD primer-extension activity. PolD was incubated at a concentration of 0.25 μM, with increasing amounts of different RPA constructs ranging from 0.05 to 0.8 μM (lanes 4-8). Lanes 1 and 2 contain an oligonucleotide ladder of 87-nt and a negative control experiment without proteins, respectively. Lane 3 contains PolD (0.25 μM) in the absence of RPA. **a** primer extension assay by PolD in the presence of RPA, **b** of Rpa2^WH, **c** of RPA^ΔWH and **d** RPA^ΔWH + Rpa2^WH. **e** Impact of RPA binding on PriSL primer-extension activity. The primer-template substrate was the same as in (**a**). PriSL (0.2 μM) was incubated with increasing amounts of different RPA constructs ranging from 0.05 to 0.8 μM (lanes 3–7). Lane 1 is the negative control without proteins. Lane 2 contains PriSL (0.2 μM) without RPA. Lane 8 contains oligonucleotide ladders (87 nt and 57-nt). From the left to the right panels: **e** primer extension assay by PriSL in the presence of RPA, **f** of Rpa2^WH, **g** of RPA^ΔWH and **h** RPA^ΔWH + Rpa2^WH. The short -20 nt PriSL primer extension products (black arrowheads) and the PolD exonuclease digestion products (*) are highlighted.

The PriSL-Rpa2[WH] complex adopts an elongated shape reminiscent of a cashew nut, as described for the heterodimeric structures of human[40,41] and *S. solfataricus*[42] primases (Supplementary Fig. 9a–c). The crystal structure shows that Rpa2[WH] binds to the PriS catalytic subunit of the DNA primase, in close proximity to the Zinc finger and AEP (archaeo-eukaryotic primase) domains[43] (Fig. 5b). The structure of the complex is in excellent agreement with the binding surface mapped from the NMR data described above (Fig. 5c). The binding surface on Rpa2[WH] is composed of $\beta_1$ and $\beta_2$ strands in the β-sheet motif, as well as the $\alpha_3$ helix. The interface is stabilized by electrostatic interactions between residue pairs R105[PriS]-E259[Rpa2], K239[PriS]-E261[Rpa2], E240/E244[PriS]-K266[Rpa2] and R317[PriS]-E250[Rpa2] (Fig. 5d, e). A representation of Coulombic electrostatic potential at the interface shows that the Rpa2[WH] interface displays a strong negative potential, and the PriS interface shows strong positive potential (Fig. 5f).

Rpa2[WH] is connected to the trimerization core of RPA by a glutamic acid-rich flexible linker (Fig. 5f). Linker residues 190–201 are visible in the electron density and were included in the final model. While no direct contact is observed between the negatively charged linker and the primase, the neighboring PriS surface is strongly positively charged. This observation is in agreement with the NMR data showing transient contacts in solution between the linker region 200–205 and PriSL[ΔCTD] (Fig. 3i–k). The primer extension assays revealed that the connection between the Rpa2[WH] and the DNA-binding core of RPA is required in order to stimulate the primer extension activity of PriSL.

When the connection between Rpa2[WH] and the RPA DNA-binding core is disrupted, not only is the stimulation of PriSL activity lost, but the addition of RPA[ΔWH] reduces synthesis of long products by the primase. To further investigate the role of this flexible, negatively charged linker and the potential importance of its acidic nature, all 12 glutamic acid residues were substituted with alanine (RPA[ala-linker]). The impact of these substitutions on the ability to stimulate PriSL primer extension activity was assessed by comparing the effects of RPA[ala-linker] versus wild-type RPA (Supplementary Fig. 10). In contrast with wild-type RPA, the RPA[ala-linker] variant showed a reduction in long extension products, ranging from 15% to 45% depending on RPA concentration, that was most prominent at low RPA concentrations. This suggests that the linker actively participates in the formation of an elongation-competent PriSL-RPA complex.

## PriSL binds to Rpa2[WH] via a canonical WH interface

In eukaryotes, Rpa2[WH] has been shown to recruit multiple repair proteins, including XPA[23,44,45], UNG2[23,26,44], RAD52[23], TIPIN[44], and SMARCAL1[24,25]. In most cases, it has been demonstrated that the WH domain recognizes a short α-helical motif within its binding partner[23–25,44]. This interaction primarily involves residues from the β-strands of the WH domain and the side chains of the target α-helical motif. Interestingly, the interface between Rpa2[WH] and the archaeal primase (Fig. 6a) resembles that of human Rpa2[WH] bound to SMARCAL1[25] (Fig. 6b), and the interface between human Stn1[WH] and

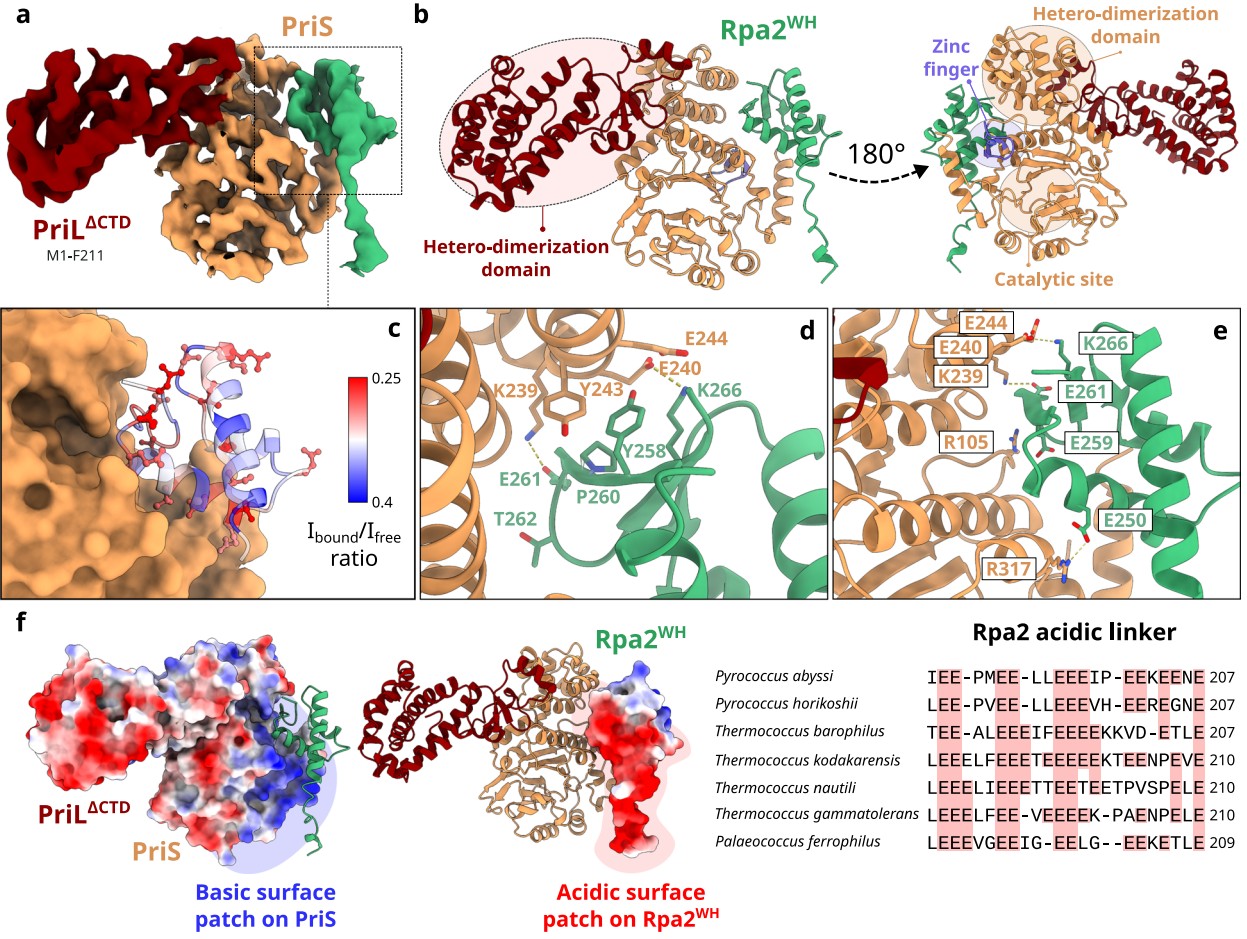

**Fig. 5 | Crystal structure of the Primase-Rpa2[WH] complex. a–c** Electron density map and model of the Primase-Rpa2[WH] complex crystal structure. The 2mFo-DFc electron density map is contoured at $\sigma = 1.5$. **c** PriS-Rpa2[WH] interface, with Rpa2[WH] residues colored according to the intensity ratio $I_{cplx}/I_{free}$ from blue (weak attenuation) to red (large attenuation) as in Fig. 3k. **d–e** Detailed view of the interface between PriS and Rpa2[WH]. **f** Surface of Primase and Rpa2[WH] colored according to their Coulomb potential, calculated in ChimeraX v1.7[84]; and multiple sequence alignment of the linker between the trimeric core helix of Rpa2 and the WH domain in *Thermococcales* showing the conservation of acidic residues.

Polα[46] (Fig. 6c). While the chemical nature of these interactions is not strictly conserved, these WH domains bind their targets through residues in the C-terminal end of helix $\alpha_3$, strand $\beta_2$, and the C-terminal end of strand $\beta_3$ in all three structures.

The Rpa2$^{WH}$-PriSL interface resembles that observed in the Stn1$^{WH}$-Polα interface in the context of the human CST telomere maintenance complex[46,47] (Fig. 6a, c). The modular architecture of Stn1 resembles that of Rpa2: Stn1 is composed of an OB domain, a trimerization helix, and two consecutive WH domains. The C-terminal WH domain (Snt1$^{WH}$) can be superimposed to Rpa2$^{WH}$ with a root mean square deviation of 2.58 Å calculated over 56 C$_\alpha$. Furthermore, although the structures of these two polymerases are different, the structures of the Rpa2$^{WH}$-PriSL and Stn1$^{WH}$-Polα complexes reveal that Polα and PriSL bind to the WH domain via a conserved canonical interface (Fig. 6c). This observation further illustrates the evolutionary kinship between the archaeal RPA and the eukaryotic CST, for which we have previously demonstrated shared features, including an AROD-like domain (Acidic Rpa1 OB-binding Domain)[22,48] and the ability to oligomerize to form supercomplexes[22,49]. We now demonstrate that they also share a similar mechanism for interacting with DNA polymerases, further establishing that archaeal RPA shares its ancestry not only with eukaryotic RPA, but also with the eukaryotic CST telomere maintenance complex.

## Molecular basis for the interaction between RPA and the replicative PolD

The structure of the Rpa2$^{WH}$–PolD complex was determined by cryo-EM at a global average resolution of 2.9 Å (Fig. 7a). The data acquisition parameters, data processing workflow and model refinement statistics are shown in Supplementary Table 3 & Supplementary Fig. 11. PolD is a heterodimeric DNA polymerase comprised of the DP1 proofreading subunit and the DP2 polymerization subunit[15,16]. The active site of PolD contains an RNA polymerase-like two-barrel catalytic core[15], surrounded by several DNA-binding domains, including Clamp-1, Clamp-2, and KH-like domains[16]. The Rpa2$^{WH}$ domain interacts with the DP2 subunit in a region that lacks any known DNA-binding domains, previously termed Accessory-1[16]. We now show that the Accessory-1 domain is involved in protein-protein interactions with other replication factors. Interestingly, the Rpa2$^{WH}$ binding site is 55 Å away from the active site of PolD (Fig. 7b, c). This contrasts with the PriSL-Rpa2$^{WH}$ binding site, which is close to its active site (Fig. 5b). The interface between Rpa2$^{WH}$ and PolD revealed in the cryo-EM map agrees with the large binding surface inferred from the HSQC experiments described above (Fig. 7b & Supplementary Fig. 8).

Surprisingly, Rpa2$^{WH}$ binds to PolD via a novel binding surface different from the canonical interface with PriSL. Indeed, the primary interactions with PolD occur predominantly within helix $\alpha_2$ and strand $\beta_1$ of Rpa2$^{WH}$ (Fig. 7f). Furthermore, a focused 3D classification step in the cryo-EM data processing workflow revealed two conformationally different populations of PolD-Rpa2$^{WH}$ particles (Fig. 7c–e). These subpopulations (named class 1 and 2) differ in a displacement of the distal side of Rpa2$^{WH}$, pivoting 7 Å around the Rpa2$^{WH}$-PolD interface (Fig. 7g, h).

Residues I474$^{DP2}$, Y475$^{DP2}$, E476$^{DP2}$, E492$^{DP2}$, Y496$^{DP2}$, V527$^{DP2}$ and R567$^{DP2}$ from an α-helical subdomain of PolD are splayed across a complementary surface of Rpa2$^{WH}$ comprising residues K229$^{Rpa2}$, Y230$^{Rpa2}$ and K233$^{Rpa2}$ from helix $\alpha_2$, residues K222$^{Rpa2}$, T224$^{Rpa2}$, S227$^{Rpa2}$ from strand $\beta_1$, and residues E261$^{Rpa2}$ and Y264$^{Rpa2}$ from the β-wing loop (Fig. 7d, e). The chemical nature of the interactions at the interface is varied and includes a network of both polar and hydrophobic contacts. In addition, two pairs of charged residues are involved in interchain electrostatic interactions: R567$^{DP2}$-E261$^{Rpa2}$, and E492$^{DP2}$-K233$^{Rpa2}$. Notably, class 2 displays an additional interaction between E492$^{DP2}$ and K233$^{Rpa2}$, bringing helix $\alpha_2$ closer to PolD and accounting for the displacement observed between class 1 and class 2.

The PolD binding surface on Rpa2$^{WH}$ is almost entirely different from the interface with PriSL, having only E261$^{Rpa2}$ in common. Therefore, we wondered whether an Rpa2$^{WH}$-Primase-PolD ternary complex could exist, as part of a polymerase-switching event. We have not been able to reconstitute such ternary complex, and superposing our Rpa2$^{WH}$-primase and Rpa2$^{WH}$-PolD experimental structures does reveal significant steric clashes between PolD and PriSL. Nevertheless, we note that the second class found in the Rpa2$^{WH}$-PolD cryo-EM dataset displays a significantly more exposed Rpa2$^{WH}$-PriSL binding interface compared to the first class (Fig. 7g, h). This conformational heterogeneity observed in the Rpa2$^{WH}$-PolD complex might allow for the primase-binding interface to remain more solvent-exposed, even if only transiently.

## Discussion

SSBs are essential components of the replisome in all three domains of life. Besides binding ssDNA, they coordinate protein-protein interactions with multiple DNA replication and repair factors in a context-dependent manner. In this study, we have characterized the protein-protein interactions between RPA and replicative polymerases in Archaea. The

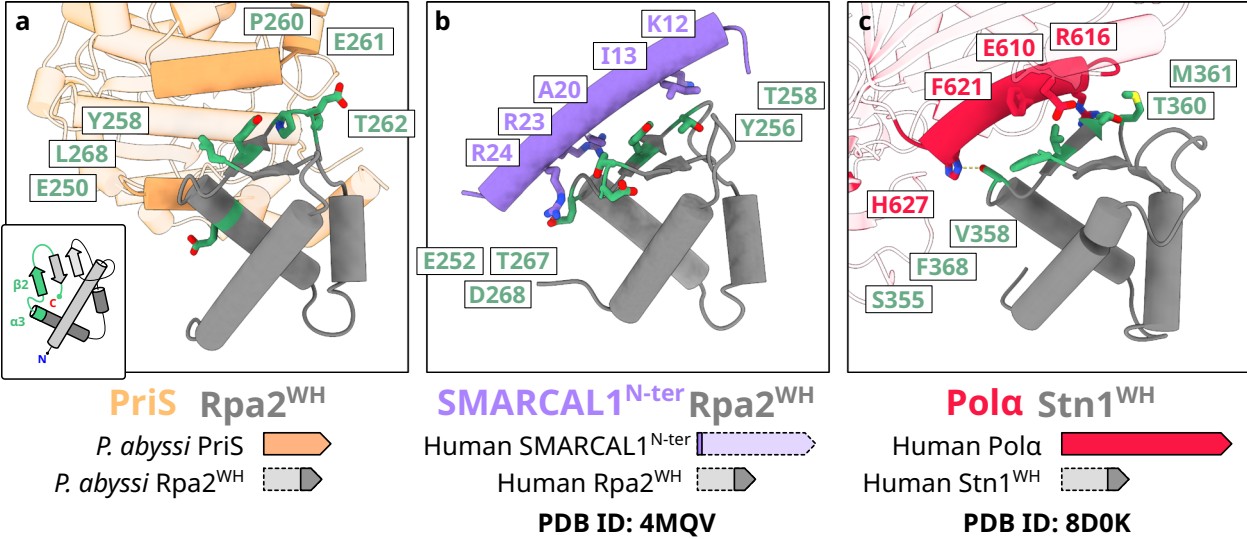

**Fig. 6 | Conserved protein-protein interaction interfaces on winged-helix domains.** Superposition of **a** *Pyrococcus abyssi* Rpa2$^{WH}$ in complex with PriS with **b** human Rpa2$^{WH}$ in complex with SMARCAL1$^{N-ter}$ (PDB ID 4MQV)[25] and **c** human Stn1$^{WH}$ in complex with Polα (PDB ID 8D0K)[46].

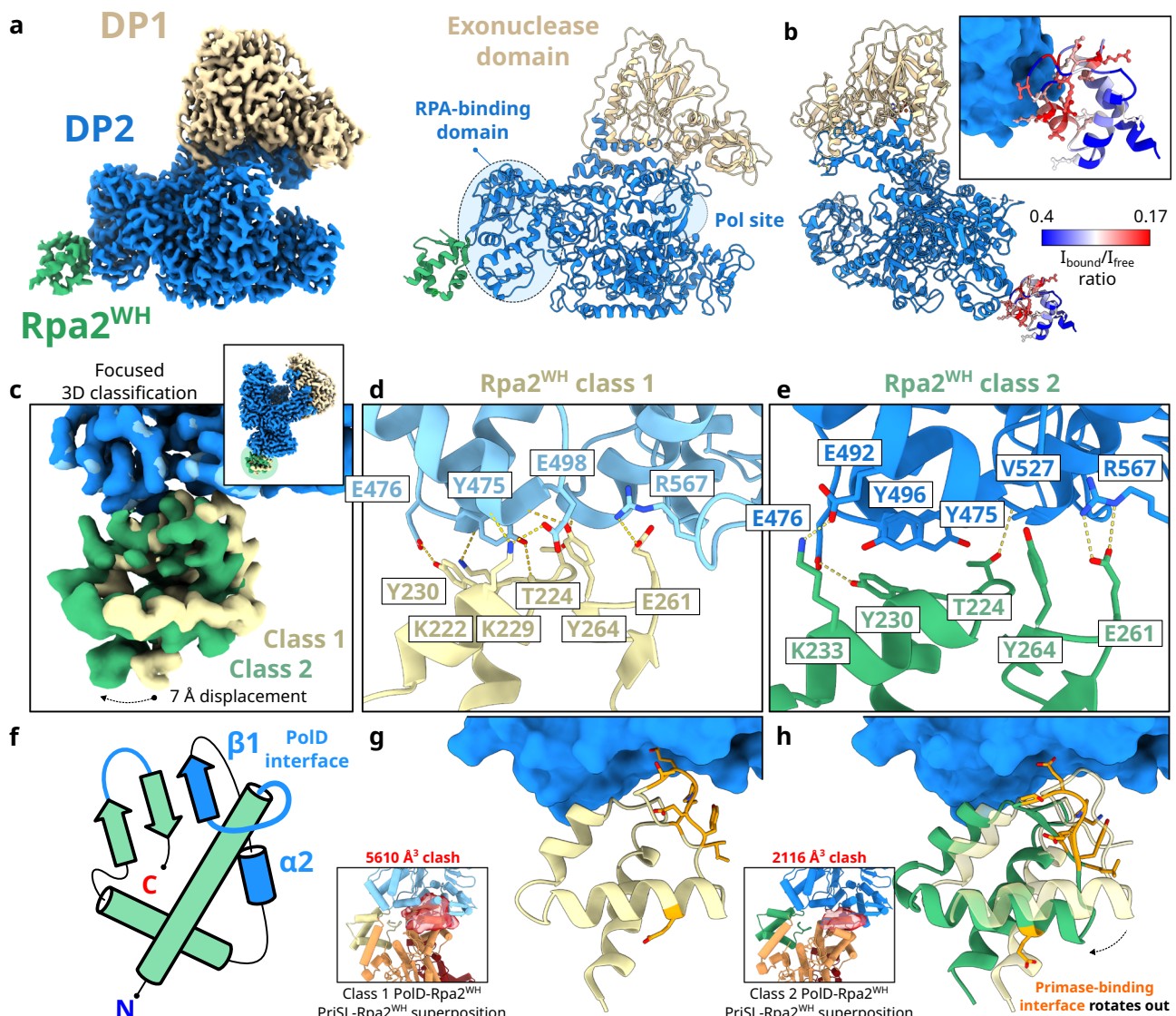

**Fig. 7 | Cryo-EM structure of PolD in complex with Rpa2^WH. a** Final composite map of PolD-Rpa2^WH at 2.9 Å global average resolution. **b** PolD-Rpa2^WH interface, with Rpa2^WH residues colored according to the intensity ratio $I_{cplx}/I_{free}$ from blue (weak attenuation) to red (large attenuation) as in Fig. 3l. **c** Superposition of the two different classes that resulted from a focused 3D classification with a soft mask around Rpa2^WH, and **d, e** a detailed view of the interface between PolD and Rpa2^WH for each class. **f** Schematic drawing of Rpa2^WH with the secondary structure elements that comprise the PolD interface colored in blue. **g, h** Comparison of the rotational displacement difference between 3D classes 1 and 2, and alignments of the Primase-Rpa2^WH structure to Rpa2^WH in the PolD-Rpa2^WH structure, showing the respective volumes of the resulting clashes in red.

biolayer interferometry assays showed that RPA-coated ssDNA has a similar affinity for PriSL and PolD compared to recombinant Rpa2^WH alone, indicating that the Rpa2 C-terminal WH domain is the main contributor to these protein-protein interactions. Notably, our genetic experiments suggest that Rpa2^WH is possibly essential in *T. barophilus*.

It is worth noting that archaeal SSBs are not uniform. Crenarchaea possess SSB proteins similar to those found in bacteria, while diverse eukaryotic-like *rpa* genes are present in Euryarchaea. The Euryarchaeon *Haloferax volcanii* has acquired multiple RPA-associated proteins (RPAPs) through gene duplication events, and its Rpa1, Rpa2 and Rpa3 do not appear to form a complex[50]. The same study found that *H. volcanii* Rpa2 is essential, and Rpa1 or Rpa3 knockouts are hypersensitive to DNA-damaging agents. On the other hand, RPA from eukaryotes and from *Thermococcales* archaeal species form hetero-trimeric complexes, and their domain arrangements are well conserved, with some noteworthy exceptions. In addition to Rpa2^WH, Eukaryotic Rpa1 contains an N-terminal OB domain (OB-F) that is also

involved in protein-protein interactions[51]. As is the case for other archaeal replication factors, RPA closely resembles its eukaryotic counterpart, but is less complex in nature, with a single WH domain in Rpa2 responsible for all protein-protein interactions identified so far. Due to its modular nature and its transient interactions with ssDNA, structural studies of RPA are often technically challenging. Using an integrative structural biology approach and building up on our previous RPA-ssDNA structural studies[22], we have characterized the RPA interactions with the archaeal replicative polymerases PriSL and PolD.

In the context of DNA replication, eukaryotic RPA has been shown to interact with Pol-α/primase in the lagging strand. Deletion of the Rfa2 WH domain, orthologous to archaeal Rpa2^WH, results in reduced interaction between RPA and Pol-α/primase[52]. More recently, the Rfa1 OB-F domain has been shown to negatively affect lagging-strand replication in vitro, while the Rfa2 WH domain positively affected it, suggesting that the two domains act in concert to regulate priming frequency in the replisome[53]. Our results indicate that the archaeal

Rpa2 WH domain interacts with PriSL as well, whose primer extension activity was stimulated by RPA in a concentration-dependent manner (Fig. 4e–h). Importantly, the stimulation was lost in reactions substituting RPA with RPA$^{\Delta WH}$, Rpa2$^{WH}$ or both RPA$^{\Delta WH}$ + Rpa2$^{WH}$, indicating that PriSL leverages its interaction with the downstream RPA through the WH domain to remain optimally engaged on the template strand and displace RPA on its way (Fig. 8b). Remarkably, as RPA$^{\Delta WH}$ acted as a roadblock for PriSL even when Rpa2$^{WH}$ was supplemented, the linker tethering the WH domain to the RPA trimeric core is crucial for PriSL to promote RPA clearance on the template strand. Interestingly, the acidic nature of this linker is conserved across *Thermococcales* (Fig. 5f), although the functional significance of the linker's amino acid composition remains to be investigated. We also considered the possibility of an allosteric stimulatory effect of Rpa2$^{WH}$ on PriSL. Comparing the crystal structures of PriSL and the PriSL-Rpa2$^{WH}$ complex, the active site in PriS does not undergo significant conformational changes upon binding of Rpa2$^{WH}$ (Supplementary Fig. 9d). Taken together with the finding that Rpa2$^{WH}$ alone does not stimulate primer elongation by PriSL (Fig. 4f). we conclude that there is no allosteric activation of PriSL by Rpa2$^{WH}$. As truncated RPA lacking the WH domain did not stimulate PriSL, our results indicate that Rpa2$^{WH}$ is required but not sufficient for primer elongation stimulation. Rpa2$^{WH}$ must also be tethered to the RPA trimeric core for the PriSL stimulatory effect to take place.

RPA has the paradoxical role of avidly coating ssDNA while selectively allowing proteins involved in DNA replication, recombination and repair to take over and perform their function[54]. In yeast,

Rad52 is a homologous recombination mediator that interacts with RPA to load the Rad51 recombinase onto RPA-coated ssDNA. Single-molecule fluorescence studies suggested that Rad52 can interact with RPA subunits Rfa1 and Rfa2$^{WH}$ resulting in a lower-footprint coating of ssDNA by RPA, thus increasing the overall accessibility of DNA[55–57]. Another study found that the protein-protein interaction domains in eukaryotic RPA can impose structural constraints on its architecture leading to intrinsic dissociation of the trimeric core from ssDNA[55]. Our results are consistent with these observations, as RPA displacement was promoted by PriSL only when they could interact through the WH domain. Our crystal structure of the Rpa2$^{WH}$-PriSL complex, validated by in-solution NMR experiments, allowed us to precisely map the PriSL-Rpa2$^{WH}$ interface and build a model of the entire PriSL-RPA interaction on DNA (Fig. 8a), aligned with our previous structure of ssDNA-bound RPA and a primer-template substrate from a PrimPol ternary complex[58]. The alignment of the structures was made minimizing the distance between the 5' and 3' ends of the template ssDNA, as well as the distance between the C-terminal of the Rpa2 core and the N-terminal of Rpa2$^{WH}$, which are only 8 residues away in the Rpa2 sequence. This model shows the relative orientation of RPA and PriSL when they interact during primer extension, indicating that RPA could optimally tether PriSL to the downstream ssDNA via the WH domain. Based on this model and our primer extension results, we have proposed a PriSL-RPA 'WH-bait' interaction model mechanism (Fig. 8b), explaining how RPA can stimulate PriSL. According to this model, PriSL will sequentially bind to the WH domain of the downstream RPA as it

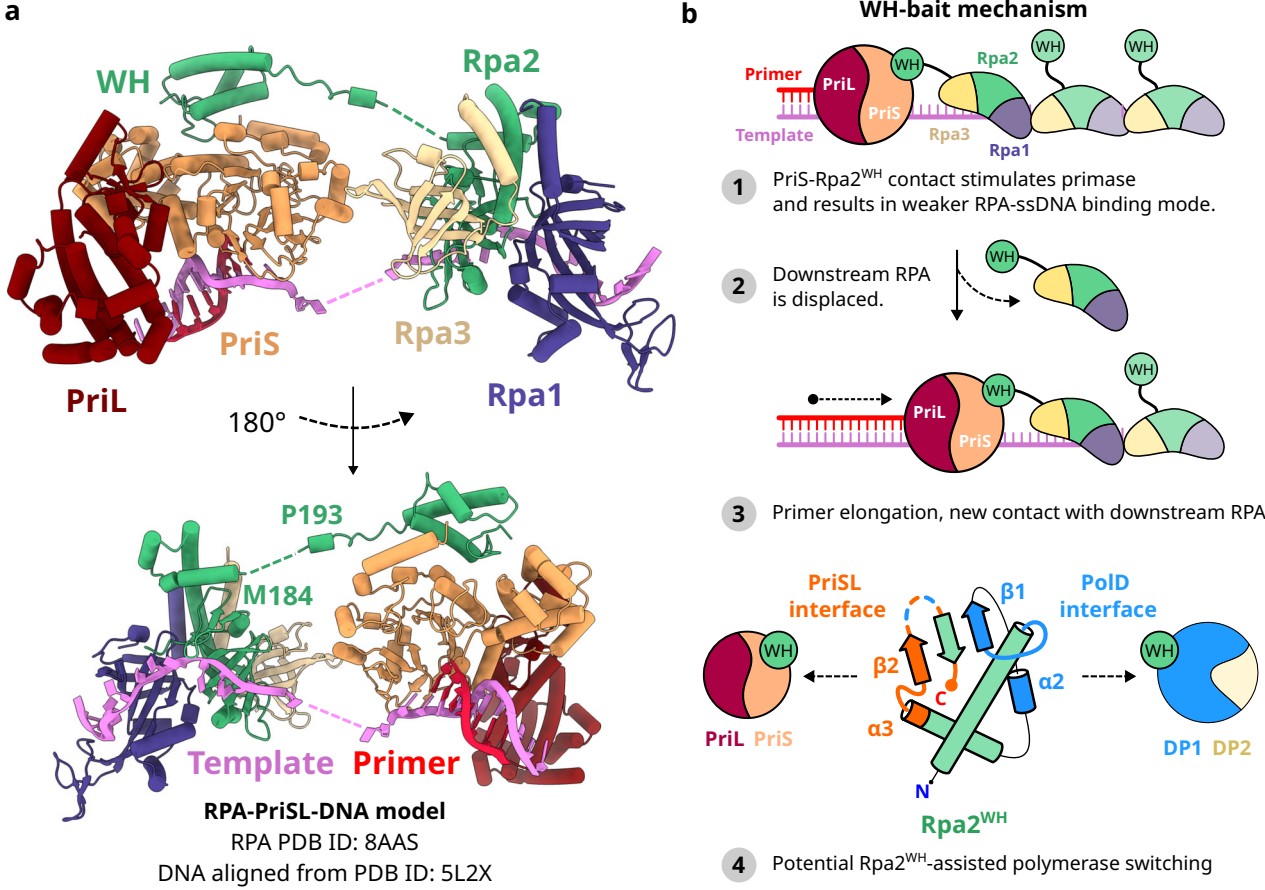

**Fig. 8 | Proposed model for the interaction of Primase and RPA during primer elongation. a** Superposition of the PriSL$^{\Delta CTD}$-Rpa2$^{WH}$ crystal structure with DNA-bound PrimPol (PDB ID 5L2X[58]), aligned to our previously reported structure of ssDNA-bound RPA trimeric core (PDB ID 8AAS[22]). The PrimPol-DNA ternary structure was used to model the DNA substrate in the PriSL active site. The RPA-ssDNA structure was oriented so that the C-terminal of the Rpa2 core would face the N-terminal of the Rpa2$^{WH}$, while matching the 5' and 3' ends of the DNA template. **b** Schematic diagram of our proposed 'WH-bait' mechanism for WH domain-dependent PriSL stimulation by RPA. A potential polymerase switch event mediated by the PolD binding site in Rpa2$^{WH}$ is shown in step 4.

elongates the primer, possibly inducing a weaker DNA-binding mode on RPA as has been observed in the Rad51-RPA interaction[56], ultimately displacing RPA from the template strand. Our study provides a structural rationale to the formerly proposed general model by which proteins trade places with RPA on ssDNA, inducing conformational changes in RPA that alter its ssDNA-binding properties[54]. After its displacement, the next downstream RPA WH domain would bind to PriSL, repeating the cycle until the primer-template hand-off to PolD takes place. We speculate that Rpa2$^{WH}$ could play a role in the DNA hand-off, as we found that RPA binds to PolD via a novel interface that remains partially solvent-accessible even as the WH domain is bound to PriSL.

Human RPA has been shown to recruit PrimPol to stalled replication forks to bypass template DNA lesions[58]. Although both PrimPol and PriSL belong to the AEP family and can perform translesion synthesis (TLS)[59–61], they interact with RPA in remarkably different manners. Notably, PrimPol binds to the OB-F domain of the upstream RPA, while PriSL binds to the WH domain of the downstream RPA. Upon binding to DNA, the catalytic subunit of PrimPol synthesizes a primer whose length is restricted by its interaction with the upstream RPA. PrimPol repriming is error-prone and tightly regulated through this mechanism, to only synthesize a short primer long enough to bypass DNA damage, as its products might contain deleterious mutations[62]. Our results indicate that in the case of PriSL, the length of the synthesis product might instead be regulated by a downstream interaction between Rpa2$^{WH}$ and PolD, or other components of the replisome.

During the submission of this article, a pre-print was published with interaction studies between human RPA and the primosome (the Pol-α/primase complex)[63]. In that study, human RPA reportedly stimulates primosome activity on DNA hairpins caused by inverted sequence repeats. Additionally, this stimulatory effect is lost in RPA mutants lacking the conserved WH domain. It is noteworthy that they found that human RPA has an inhibitory effect on primosome activity on unstructured DNA. Nevertheless, and in agreement with our findings on archaeal RPA, this inhibitory effect was increased in RPA mutants lacking the WH domain.

The interface between Rpa2$^{WH}$ and PriSL is well conserved in human Rpa2$^{WH}$ and Stn1$^{WH}$, as shown in experimental structures with their respective interaction partners SMARCAL1 and Pol-α[25] (Fig. 6). Surprisingly, our PolD-Rpa2$^{WH}$ cryo-EM structure and NMR data revealed that a domain as small as the 62 amino acid-long Rpa2$^{WH}$ contains a second binding site that has never been found in other WH domains, hosting the interface to the archaeal replicative polymerase PolD (Fig. 7f). The structure and interacting residues of the WH domain at the PolD interface are well conserved within *Thermococcales* but not in eukaryotes, which is expected, as PolD does not exist in eukaryotes (Supplementary Fig. 12). The binding site on PolD is in a region that had no previously described function, termed the Accessory-1 domain. Contrarily to its effect on PriSL, RPA acted as a roadblock for PolD in our primer extension assays in a concentration-dependent manner (Fig. 4a). Nevertheless, we previously showed that PCNA enhances the processivity and exonuclease activity of PolD on DNA mispairs, while RPA does not severely affect this specific activity[36,64]. It is likely that the PolD-PCNA complex exhibits a more intricate behavior interacting with RPA-coated ssDNA in the replisome.

The interaction we observe between PolD and RPA could occur in different settings in the context of cellular DNA replication, as the replisome is composed of multiple replication factors that dynamically interact with each other to simultaneously carry out leading and lagging-strand synthesis. Studies mutating the interface between bacterial Pol III and SSB have shown that the interaction between the Pol III χ subunit and SSB is important to enhance the stability of the entire replisome[65]. Additionally, eukaryotic RPA enhances the unwinding efficiency of the CMG replicative helicase through a mechanism beyond its ability to prevent strand re-annealing[53]. As PolD performs DNA replication in both leading and lagging strands, the PolD-RPA interaction is also likely to play an important role in the stability of the archaeal replisome. Further insights into the RPA-PolD interaction will be needed to understand its role in the replisome, with non-lethal mutants of the Rpa2-PolD interface or with single-molecule studies to track potential polymerase switch events in the presence of RPA.

We found that Rpa2$^{WH}$ binds to PolD with relatively high affinity ($K_D = 98 \pm 49$ nM), yet the cryo-EM dataset showed that this interaction is not uniformly tight, as we found two distinct subpopulations of particles where the position of Rpa2$^{WH}$ differs by a pivoting rotation of ~7 Å around the interface (Fig. 7c–h). Superposing Rpa2$^{WH}$ from the PriSL-Rpa2$^{WH}$ crystal structure with the PolD-Rpa2$^{WH}$ cryo-EM structure leads to severe steric clashes between the polymerases. However, the position of Rpa2$^{WH}$ in class 2 exposes the PriSL-binding interface to the solvent to a greater degree, reducing the volume of the superposition steric clash from 5610 Å³ to 2116 Å³ (Fig. 7g, h). Therefore, we speculate that since both complexes seem only partially mutually exclusive, RPA could potentially play a role in a polymerase-switch event between PriSL and PolD. It is noteworthy that the PriSL-Rpa2$^{WH}$ complex, when superposed with the PolD-Rpa2$^{WH}$ complex, would have to transfer the dsDNA to the active site of PolD over 90 Å away (Fig. 7a). This suggests that the entire hand-off process is likely to also be driven by additional molecular interactions, either directly between PriSL and PolD or with the contribution of additional replication or repair factors. Investigating this transient potentially RPA-mediated polymerase switch will be crucial to understand the dynamic nature of the archaeal replisome.

In Archaea, the polymerase-switch event could in principle be bidirectional. The crenarchaeal replicative primase PriSLX has recently been reported to initiate priming by synthesizing an RNA primer, to later stochastically start incorporating dNTPs during elongation, which facilitates dsDNA hand-off to the replicative polymerase[37]. On the other hand, we and others have previously shown that PolD and PolB stall at damaged 8-oxodG-containing template sites, where PriSL can take over to bypass the damage performing translesion synthesis (TLS), stimulated by both PCNA and RPA[36,60]. Our current study describes how RPA interacts with the archaeal DNA polymerases, and indicates that RPA is likely to play a role in the regulation of the switching between PriSL and PolD. We also identified Rpa2$^{WH}$ as the main contributor to protein-protein interactions by archaeal RPA, and observed that this domain stimulates long primer extension by PriSL. Additionally, using an integrative approach combining cryo-EM, X-ray crystallography and NMR, we determined the structures of Rpa2$^{WH}$ bound to PriSL and PolD, revealing a novel interface with PolD in Rpa2$^{WH}$ besides the canonical interface with PriSL.

Collectively, our results show that RPA-mediated protein-protein interactions are functionally essential in the archaeal replisome. It is of great interest to understand how the replisome regulates the activity of its multiple replication factors to dynamically and contextually orchestrate which protein should interact with RPA-coated ssDNA. Further structural insights into hub replication factors such as RPA will be crucial to better understand the complex mechanisms governing genome maintenance.

## Methods

### Cloning, protein expression, and purification

The open reading frames (ORFs) of the Rpa1, Rpa2, Rpa2$^{ala-linker}$, Rpa3, PriS, PriL, PolD$^{DP1}$ and PolD$^{DP2}$ genes from *P. abyssi* were optimized and synthesized by GeneArt (ThermoFisher). For individual subunit expressions, ORFs were inserted into the pRSFduet(+) (Novagen) multiple cloning site 1 with a TEV-cleavable N-terminal 14-His tag[15]. For PriSL, ORFs were inserted into the pRSFduet(+) multiple cloning site 1 as a polycistronic PriS-PriL construct separated by ribosome binding sites (RBS), with a TEV-cleavable N-terminal His14-tagged PriS fusion protein. The PriS-PriL$^{ΔCTD}$-Rpa2$^{WH}$ complex was generated by cloning a polycistronic Rpa2$_{(190-268)}$ construct into the multicloning site 1 and

PriS-PriL$_{(1–210)}$ into the multicloning site 2. PolD and PriS-PriL$^{\Delta CTD}$ were cloned as previously described[16,64]. For RPA complexes co-expressions, ORFs were cloned in a polycistronic construct as previously described[22]. For Rpa$^{ala-linker}$, the Rpa2 linker residues E188–189, E192–193, E196–198, E201–202, E204–205 and E207 were substituted into alanines. RPA isoforms RPA$^{\Delta WH}$ (Rpa1/Rpa2$_{1-179}$/Rpa3) and Rpa2$^{WH}$ (Rpa2$_{178-268}$) were cloned from the pRSFduet(+) constructs using the Q5 site directed mutagenesis kit (E0554, New England Biolabs).

Proteins were expressed in BL21 Star (DE3) strain from *Escherichia coli* (Invitrogen) at 37 °C in Luria–Bertani (LB) medium supplemented with 100 µg/mL of kanamycin. For Rpa2$^{WH}$ $^{15}$N and $^{13}$C uniformly labeled samples used in NMR, cells were grown in minimal media supplemented with yeast nitrogen base without amino acids and ammonium sulfate (DIFCO) and containing $^{13}$C$_6$ glucose and $^{15}$NH$_4$Cl (Eurisotop) as sole sources of carbon and nitrogen, respectively. Recombinant protein expression was induced by adding 0.25 mM IPTG (0008-B Euromedex). Cells were then incubated overnight at 20 °C, collected by centrifugation, resuspended in buffer A (0.02 M Na-HEPES (H7006 Sigma) at pH 8, 0.5 M NaCl (S9888 Sigma), 0.02 M imidazole (I202 Sigma)) supplemented with complete EDTA-free protease inhibitors (Roche), and lysed with a Cell-Disruptor (constant systems LTD, Northants, UK). Lysates were heated for 10 min at 60 °C and centrifuged 30 min at 20,000 × *g*. PriSL and PriS-PriL$^{\Delta CTD}$ purifications were performed using a three-step protocol including nickel affinity, heparin affinity and size exclusion chromatography. The clear cell lysate was loaded onto 5 mL HisTrap columns (17528601 Cytiva) connected to an ÄKTA purifier (Cytiva). Elutions were performed using a linear gradient of imidazole (buffer B, 0.02 M Na-HEPES at pH 8, 0.5 M NaCl, 0.5 M imidazole). Protein fractions were combined, dialyzed in buffer C (0.02 M Na-HEPES pH 8, 0.1 M NaCl), loaded onto 5 ml HiTrap Heparin columns (17040703 Cytiva) and eluted with a linear gradient, by mixing buffer C with buffer D (0.02 M Na-HEPES pH 8, 2 M NaCl). Depending on the applications, the 14-His tag was removed following an overnight TEV-protease cleavage. Purifications were finally polished using exclusion-size chromatography in buffer E (0.02 M Na-Hepes pH 8, 0,15 M NaCl) on a Superdex 200 10/300 (Cytiva). Rpa2$^{WH}$ and the PriS-PriL$^{\Delta CTD}$-Rpa2$^{WH}$ complex were purified following the same protocol without the anion exchange chromatography step. PolD, RPA and RPA$^{\Delta WH}$ were purified as previously described[22,64]. Briefly, all three constructs were expressed in *E. coli* BL21 Star (DE3) competent cells, which were harvested and lysed as the other constructs described above. After gradient elution from the HisTrap, PolD fractions were combined and dialyzed in buffer C (0.02 M Na-HEPES pH 8, 0.1 M NaCl), then loaded on a HiTrap Heparin 5 ml column and eluted with a linear gradient mixing buffer C with buffer D (0.02 M Na-HEPES pH 8, 2 M NaCl). RPA and RPA$^{\Delta WH}$ were purified similarly, but using a HiTrap Q 5 ml column instead of the Heparin column. All three constructs were further purified by size-exclusion chromatography in an S200 10/300 column in buffer E (0.02 M Na-Hepes pH 8, 0,15 M NaCl).

## Crystallization, X-ray data collection, and processing

Crystallization conditions were identified after extensive screening by the crystallization platform of the Pasteur Institute[66]. Crystallization trials were performed at 18 °C using the hanging drop vapor diffusion technique in 2 µL drops (1:1 reservoir to protein ratio) equilibrated against 500 µL of reservoir solution. PriS-PriL$_{(1–210)}$ crystals were obtained in 0.2 M magnesium formate with a protein solution at 30 mg/mL, whereas PriS-PriL$_{(1–210)}$-Rpa2$_{(190-268)}$ crystals were obtained in 0.1 M Bis-Tris pH 7 60 %v/v Tacsimate at 10 mg mL$^{-1}$. The crystals were cryoprotected with 25% ethylene glycol. X-ray data were collected at the SOLEIL synchrotron on beamlines PX1 and PX2. Datasets were indexed using XDS, scaled and merged with Aimless (from the CCP4 program suite (Collaborative Computational Project 1994)[67], and

corrected for anisotropy with the STARANISO server (star-aniso.globalphasing.org). PriS-PriL$_{(1–210)}$ X-ray structure was solved by molecular replacement using the structure of primase from *S. solfataricus* (PDB ID: 1ZTD). The crystal structure of the PriS-PriL$_{(1–210)}$-Rpa2$_{(190-268)}$ complex was determined by molecular replacement with the PriS-PriL$_{(1–210)}$ X-ray structure and the Rpa2$^{WH}$ domain NMR structure as initial models. Molecular replacement was carried out with the Phaser program from Phenix[68] and subsequent rebuilding and refinement were achieved with COOT[69] and BUSTER[70].

## NMR

NMR experiments were performed on Bruker Avance Neo 800 MHz or Avance III HD 600 MHz spectrometers (Billerica, USA) with 18.8 and 14.1 T magnetic fields, respectively. Both spectrometers were equipped with cryogenically cooled triple resonance ($^1$H/$^{13}$C)/$^{15}$N TCI probes. Data were acquired with Topspin 4.1.3 or 3.6.5 (Bruker), processed with Topspin and NMRPipe[71], and analyzed with CCPNMR analysis 2.5.2[72].

Experiments were performed at 35 °C with $^{15}$N or $^{15}$N/$^{13}$C labeled Rpa2$^{WH}$ (300 µM) samples prepared in 20 mM MES pH 6, 150 mM NaCl 5% D$_2$O for assignment, structure calculations and $^{15}$N relaxation measurements.

Backbone and side chain $^1$H/$^{15}$N/$^{13}$C assignments were obtained by standard methods, from $^1$H-$^{15}$N HSQC, $^1$H-$^{13}$C HSQC, HCCH-TOCSY and 3D BEST versions of HN(CO)CACB, HNCACB, HNCA, HN(CO)CA implemented in NMRLib 2.0[73], C(CO)NH-TOCSY, $^1$H-$^{15}$N HSQC-TOCSY, HBHAN, HBHA(CO)NH. Aromatic $^1$H/$^{13}$C spin systems were established from aromatic $^1$H-$^{13}$C HSQC CDHE experiments and were sequentially assigned through aromatic-aliphatic NOEs in $^{15}$N- and $^{13}$C-edited HSQC-NOESY experiments.

NOE assignment and structure calculations were performed using ARIA 2.3.2[74] and CNS 1.2.1[75]. NOEs from $^1$H–$^{15}$N HSQC-NOESY and $^1$H–$^{13}$C HSQC-NOESY (with 150 and 120 ms mixing time, respectively) obtained on the 800 MHz spectrometer were used to derive distance constraints. Phi and Psi dihedral angle constraints were obtained from backbone and CB chemical shifts using TALOS-N[76]. Additionally, hydrogen bond constraints were used when in agreement with the pattern of NOEs expected for regular α-helix and β-sheet secondary structures. Calculations were performed using the log-harmonic potential, network anchoring and spin-diffusion correction as implemented in ARIA, considering a rotational correlation time of 5.7 ns obtained from $^1$H-$^{15}$N TRACT[77] experiments. To obtain the final structural ensemble, 200 structures were calculated and refined in explicit water. The best 10 structures in terms of total energy were selected. Secondary structures were determined from the structural ensemble with DSSP[78].

The internal dynamics of Rpa2$^{WH}$ were analyzed from $^{15}$N relaxation measurements performed on the 600 MHz spectrometer. The $^{15}$N relaxation rates ($R_1$ and $R_2$) and {$^1$H}-$^{15}$N heteronuclear NOE were recorded by standard methods implemented in NMRLib 2.0[73], in an interleaved manner with a recycling time of 3 s and with nine relaxation delays for $R_1$ (20, 100, 200, 350, 500, 700, 950, 1300, 2000 ms) and eleven for $R_2$ (0, 17, 34, 68, 102, 153, 220, 305, 509, 848, 1272 ms). The heteronuclear NOE were recorded in the presence and absence of a 3 s $^1$H saturation sequence (120° $^1$H pulse train). The relaxation parameters were analyzed with the program TENSOR2[79] to infer global and internal motions. To describe the global reorientation of the globular domain (E206-L268), an isotropic model with a correlation time $\tau_c$ of 5.5 ns estimated from the $R_2/R_1$ ratios of non-flexible residues (in agreement with the value of 5.7 ns obtained from TRACT experiments) could not correctly fit the relaxation data. Due to the non-spherical shape of the domain, the impact of the N-terminal disordered tail and the profile of the $R_2/R_1$ ratios (Supplementary Fig. 2), a fully anisotropic model (with a diffusion tensor Dx, Dy, Dz (1e$^7$ s$^{-1}$) of 2.24 ± 0.05, 2.62 ± 0.06, 4.38 ± 0.06) was used, which clearly improved the fit. The relaxation parameters were then analyzed using the Lipari and Szabo formalism[80] to extract internal dynamical parameters (order parameter $S^2$, internal

correlation time $\tau_e$ on the ps-ns timescale and exchange parameter $R_{ex}$ on the µs-ms timescale).

Interactions of $^{15}N$-labeled Rpa2$^{WH}$ with unlabeled PriSL$^{\Delta CTD}$, PolD and PolB were probed at 35 °C in 20 mM HEPES pH 7.5, 200 mM NaCl 5 mM MgCl$_2$ 5% D$_2$O. $^1H$-$^{15}N$ HSQC were recorded on 100 µM $^{15}N$-labeled Rpa2$^{WH}$ in the absence and presence of 50 µM unlabeled PriSL$^{\Delta CTD}$ (ratio 1:0.5) or 10 µM unlabeled PolD (ratio 1:0.1). For PolB, 50 µM $^{15}N$-labeled Rpa2$^{WH}$ in the absence and presence of 50 µM unlabeled PolB (ratio 1:1) were used. Chemical Shift Perturbations (CSP) on Rpa2$^{WH}$ induced by the presence of PriSL$^{\Delta CTD}$, PolD or PolB were calculated as the weighted average ($^1H$, $^{15}N$) chemical shift differences ($\Delta\delta$) between free and bound forms as follows: CSP = $((\Delta\delta(^1H))^2 + (\Delta\delta(^{15}N) \times 0.159)^2)^{1/2}$. Intensity ratios $I_{cplx}/I_{free}$ are calculated from Rpa2$^{WH}$ peak intensities in the complexed ($I_{cplx}$) and free ($I_{free}$) forms. Errors on intensity ratios are determined as: $\Delta(I_{cplx}/I_{free}) = I_{cplx}/I_{free} \times ((\Delta I_{cplx}/I_{cplx})^2 + (\Delta I_{ref}/I_{ref})^2)^{1/2}$, where $\Delta I_{cplx}$ and $\Delta I_{free}$ represent the noise standard deviation in the spectra of complexed and free forms of Rpa2$^{WH}$.

### Cryo-EM sample preparation

The PolD-Rpa2$^{WH}$ complex was obtained by injecting previously purified and concentrated PolD in a Superdex 200 10/300 size exclusion chromatography column equilibrated in 20 mM Tris-HCl pH 8.0, 150 mM NaCl, 5 mM MgCl$_2$, 1 mM DTT, 0.02% NP40). The resulting peak was concentrated to 6 mg/ml, and Rpa2$^{WH}$ was added in a fivefold molar excess. Grids were frozen in a Vitrobot Mark IV (ThermoFisher Scientific), applying 3 µl of sample onto previously glow-discharged Quantifoil R2/2 300 copper mesh grids (Electron Microscopy Sciences). Grids were blotted for 4 s at 100% humidity and 22 °C prior to being vitrified in liquid ethane.

### Cryo-EM data collection and processing

The PolD-Rpa2$^{WH}$ complex reconstruction was obtained from a dataset collected on a Titan Krios G31 operating at 300 kV and equipped with a Falcon 4 direct electron detector and a Selectris X energy filter at the OPIC electron microscopy facility (Oxford, UK Instruct-ERIC Centre). 8,371 movies were collected at 40 e$^-$/Å$^2$ flux and 130,000X magnification (0.96 Å/px). Additional data collection parameters are specified in Supplementary Table 3.

Data processing was carried out in Cryosparc v4.1[81]. The overall workflow for data processing is depicted in Supplementary Fig. 8 Patch-motion correction and patch-CTF estimation were performed on the raw movies, followed by preliminary blob-picking in a random subset of micrographs. Blob picks were filtered by 2D classification to find a subset of particles to train a Topaz picking model[82]. Topaz particle picking in the whole set of micrographs resulted in 605,481 particles, which were used in an ab-initio reconstruction step (two classes) and subsequent heterogeneous refinement. The map of the resulting class that properly resembled the PolD-Rpa2$^{WH}$ complex revealed discernible heterogeneity in the region of Rpa2$^{WH}$, which was best resolved with 3D classification using a soft focus mask around Rpa2$^{WH}$. The resulting three 3D classes showed PolD alone and two distinct conformations of Rpa2$^{WH}$, respectively. The latter two were named 3D classes 1 and 2. As the DP1 domain of PolD was visibly blurred in the map, local refinement of this region was performed individually for each 3D class. The individual components of each map were processed in DeepEMhancer[83], and final composite map reconstructions of 3D classes 1 and 2 were performed in ChimeraX[84].

### M13-templated primase assay

Reactions (20 µL) were carried out in buffer (25 mM HEPES pH 6.8, 1 mM DTT, 5 mM MnCl$_2$, 10 mM MgCl$_2$, 100 mM NaCl) containing 4.2 pmol M13mp18 ssDNA, 1 µM PriSL and all eight nucleotides at physiological concentrations (95 µM dATP, 103 µM dGTP, 200 µM dTTP,

33 µM dCTP, 3359 µM rATP, 2157 µM rGTP, 1889 µM rUTP and 981 µM rCTP) as determined previously. Primase assays were also performed in the presence of all four dNTPs or all four rNTPs using the concentrations detailed above. Individual reactions were pre-incubated at 60 °C for 10 min with increasing concentrations of RPA (0, 0.2, 0.4, 0.8, 1.6, 3.2, 6.4 µM) before addition of PriSL. After 1 h at 60 °C, reactions were quenched on ice by adding an equal volume of stop buffer (98% deionized formamide and 25 mM EDTA) and boiled at 90 °C for 3 min. Products (20 µL) were resolved on a 2% (w/v) agarose gel 1 × TBE gel. Electrophoresis was performed at 4 °C for 14h30 at 30 V followed by SYBR Gold gel staining for 30 min. Products were visualized on a Typhoon FLA 9500 Imager (GE Healthcare, Life Sciences). 1 kb Plus DNA Ladder NEB (N3200S) and Low Range ssRNA Ladder NEB (N0364S) were used as DNA and RNA markers, respectively.

### Primer extension assay

Reactions (10 µL) were carried out in buffer (25 mM HEPES pH 6.8, 1 mM DTT, 5 mM MnCl$_2$, 10 mM MgCl$_2$, 100 mM NaCl) containing 50 nM fluorescently labeled primer annealed to the oligodeoxynucleotide DNA template (5′-Cy5 P17/T87, Supplementary Table 4), 200 nM PriSL or 250 nM PolD and all eight nucleotides at physiological concentrations (95 µM dATP, 103 µM dGTP, 200 µM dTTP, 33 µM dCTP, 3359 µM rATP, 2157 µM rGTP, 1889 µM rUTP and 981 µM rCTP). Individual reactions were pre-incubated at 55 °C for 10 min with increasing concentrations of RPA, RPA$^{\Delta WH}$, Rpa2$^{WH}$, or RPA$^{\Delta WH}$ + Rpa2$^{WH}$ (0, 0.05, 0.1, 0.2, 0.4, 0.8 µM) before addition of PriSL or PolD. After 1 hour at 55 °C, reactions were quenched on ice by adding an equal volume of stop buffer [98% deionized formamide, 10 mM NaOH, 10 mM EDTA (pH 8), 1 µM 《 oligonucleotide competitor 》 (the exact complement of the template strand under study)] and samples were heated at 90 °C for 5 min, before being loaded onto 17% polyacrylamide, 8 M urea, 1 × TBE gel. Electrophoresis was performed for 4h30 hours at 5 W. Labeled products were detected with Typhoon FLA 9500 Imager and quantified with ImageQuant TL 8.1 software. PriSL extension products were quantified calculating the intensity of products ≥70 nt in length as a percentage of total lane intensity. For extension product size determination, standard curves of the log MW (molecular weight) *versus* relative migration distance (Rf) were generated using the ladder 87-nt, 57-nt and 17-nt bands. Unknown size products were then calculated according to the linear curve. PolD extension products were quantified measuring the intensity of the bands of full-length products (87 nt in length) as a percentage of total lane intensity. Full-length products are determined according to the migration position of the FAM-labeled oligonucleotide ladder 87 nt-in length (the exact sequence template synthesized under study).

### Ribonucleotide detection in dsDNA

Primer extension reactions (20 µL) were performed as described above and quenched on ice by addition of EDTA (18 mM final concentration). Half of the reaction mixture (10 µl) was incubated at 55 °C for 2 h in either 0.25 NaOH or 0.25 NaCl. Reactions were quenched on ice by adding an equal volume of stop buffer [98% deionized formamide, 10 mM NaOH, 10 mM EDTA (pH 8), 1 µM 《 oligonucleotide competitor 》 (the exact complement of the template strand under study)] and samples were heated at 90 °C for 3 min, before being loaded onto 17% polyacrylamide, 8 M urea, 1 × TBE gel. Electrophoresis was performed for 4h30 hours at 5 W. Labeled products were detected with Typhoon FLA 9500 Imager and quantified with ImageQuant TL 8.1 software as described above. As control, 50 nM of fluorescently labeled double-stranded DNA containing a single embedded ribonucleotide was submitted to either NaOH or NaCl treatment using the same procedure as for primer extension products.

## Construction of *Thermococcus barophilus* and derivative mutant strains

**Strains, media, and growth conditions.** Bacterial and archaeal strains are listed in Supplementary Table 5. *E. coli* strain DH5α was the general cloning host. LB broth was used to cultivate *E. coli*. *Thermococcales* rich medium (TRM) was used to cultivate *T. barophilus*, under anaerobic condition and at 85 °C, as described by Zeng et al.[85]. Media were supplemented with the appropriate antibiotics used at the following concentrations: kanamycin 50 μg/ml and ampicillin 100 μg/ml for *E. coli*, simvastatin 2.5 μg/ml for *T. barophilus*. When necessary, elemental or colloidal sulfur (0.1 % final concentration) was added for *T. barophilus*. Plating was performed by addition to the liquid medium of 16 g/l of agar for *E. coli* and 10 g/l of phytagel for *T. barophilus*.

**Transformation methods.** The transformation of *T. barophilus* was performed as previously described[86] using 0.2 to 2 μg of plasmid.

**Construction of plasmids.** Most of the constructions were inserted into pUPH[32] using *Kpn*I/*Bam*HI restriction sites. A list of primers is given in Supplementary Table 6. The plasmid to construct the deletion of the C-terminal of RPA32 (RPA32ΔWH; *TERMP_01998*) was constructed using the fusion of three DNA fragments obtained previously with primers pair 761-RPA32ΔWHCterKpnI/762-RPA32ΔWHCterFusRv, 763-RPA32Δ70AACterFusFw/772-RPA-SupBamHI-Rv and 771-RPA-Sup-BamHI-Fw/764-RPA32ΔWHCterBamHI. The fusion was done using 764-RPA32ΔWHCterBamHI/761-RPA32ΔWHCterKpnI. After cloning into pUPH, plasmid was named pRD603. Potential mutant of *T. barophilus* were analyzed using 766-RPA32ΔWHVerifRv/776-RPAΔ3SUVerifFw primer pair. The plasmid to construct mutant of the three *RPA* (*TERMP_01996*, *TERMP_01997* and *TERMP_01998*) was constructed using the fusion of two DNA fragments obtained previously with primers pair 773-RPAΔ3SU-KpnI/775-RPAΔ3SU-FusFw and 774-RPAΔ3SU-FusRv/764-RPA32ΔWHCterBamHI. The fusion was done using 764-RPA32ΔWHCterBamHI/773-RPAΔ3SU-KpnI. After cloning into pUPH, plasmid was named pRD605. Potential mutant of *T. barophilus* were analyzed using 766-RPA32ΔWHVerifRv/776-RPAΔ3SUVerifFw primer pair.

## Biolayer Interferometry specific protein–protein interaction assays

Protein-protein interaction assays were performed on an Octet RED384 BLI instrument (ForteBio). Each binding experiment was performed at least two times at 25 °C in buffer E supplemented with 0.2 mg/ml BSA. Protein baits were immobilized by using histidine-tagged constructs captures with Ni-NTA sensors, and ssDNA baits were immobilized capturing a 3′-TEG-Biotin labeled (CACGCCCTACCTC-CATGATCCACTGACCTCCCAGACGCTGCAAGACTTCC) oligonucleotide on streptavidin sensors. Steady-state analysis was performed by subtracting the signal from a reference well without protein in the association phase from all sensors, and fitting the function Req = Rmax*[Protein]/($K_D$ + [Protein]) to the mean and standard deviation values obtained from the triplicate assays. The concentration and constructs used for each experiment is summarized in the legend of Fig. 3.

## Reporting summary

Further information on research design is available in the Nature Portfolio Reporting Summary linked to this article.

## Data availability

Cryo-EM: Individual component maps of the DP1 domain and DP2-Rpa2WH from 3D class 1 are available on the EMDB as entries EMD-50141 and EMD-50142, respectively. Individual component maps of the DP1 domain and DP2-Rpa2WH from 3D class 2 are available on the EMDB as entries EMD-50144 and EMD-50145, respectively. Coordinates and composite maps of 3D classes 1 and 2 are available as PDB entries 9F29 and 9F2A and EMDB entries EMD-50140 and EMD-50143, respectively. NMR: Coordinates and chemical shifts of Rpa2WH are deposited on the PDB with the accession code 9F27 and on the BMRB with the accession code BMRB 34913, respectively. X-ray crystallography: Coordinates and structure factors of the PriS-PriL$_{(1–210)}$ and PriS-PriL$_{(1–210)}$-RPA2$_{(190–268)}$ crystal structures from *P. abyssi* were deposited in the Protein Data Bank under accession codes 9F28 and 9F26, respectively. Source data are provided with this paper.

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

## Acknowledgements

We gratefully acknowledge the financial support of ANR (Grant ARCH-APRIM). C.M. was funded by a postdoctoral Pasteur-Roux-Cantarini fellowship. M.M.C. was funded by a postdoctoral FRM fellowship (Fondation pour la Recherche Médicale). We wish to acknowledge SOLEIL for the provision of synchrotron radiation facilities and the staff of beamlines PROXIMA-1 and PROXIMA-2 (Saint-Aubin, France). We would like to thank the staff of the Nanoimaging Core Facility at Institut Pasteur for cryo-EM instrument access for the preliminary cryo-EM datasets. Electron microscopy instrument time was provided through the OPIC electron microscopy facility, a UK Instruct-ERIC Centre, which was founded by a Welcome JIF award (060208/Z/00/Z) and is supported by a Welcome equipment grant (093305/Z/10/Z). Cryo-EM final data collection experiments were funded by UK Instruct-ERIC, PID: 21658. The 800-MHz NMR spectrometer of the Institut Pasteur was partially funded by the Région Ile de France (SESAME 2014 NMRCHR grant no 4014526).

## Author contributions

M.M.C., L.V., C.M., F.C., A.D.T., A.H., P.L., R.A.L., P.E., R.D., J.I.G, G.H. and L.S. conceived the experiments, and analyzed the data. C.M. performed molecular cloning. C.M., A.D.T., and M.M.C. performed protein expression and purification. M.M.C. prepared the cryo-EM grids, analyzed the data, and reconstructed the cryo-EM maps. C.M., L.S and A.H. prepared the crystals, and P.L. and C.M. collected the X-ray diffraction data and determined the crystal structures. P.L., C.M. and L.S. refined and deposited the crystal structures. L.V. and G.H. performed the primer extension and priming activity assays. F.C. R.A.L. and J.I.G. collected the NMR data, built the model and analyzed the NMR data. L.V, R.D. and G.H. performed all the genetic studies. M.M.C, C.M. and P.E. performed all BLI assays. M.M.C., C.M., F.C., L.V., J.I.G., G.H. and L.S. elaborated and prepared the figures. M.M.C., L.V., C.M., F.C., A.H., P.E., P.L., R.D., J.I.G., G.H., and L.S. made suggestions and approved the final manuscript. G.H. and L.S. supervised the project.

## Competing interests

The authors declare no competing interests.
