## [Transparent Peer Review file · Nature Communications]

Communication between DNA polymerases and Replication Protein A within the archaeal replisome

Corresponding Author: Dr Ludovic Sauguet

Version 0:

Reviewer comments:

Reviewer #1

(Remarks to the Author)
Review

The authors describe structurally how archaeal RPA2 binds PriSL, archaeal primase, and PolD, the replicative polymerase, as a means to better understand the communication between polymerases and RPA in the replisome, as stated in the title. The WH domain of RPA2 is identified as the main interaction site. The authors then use in vitro analysis to characterise the effect of the critical binding determinants in the PRA2-WH domain on PriSL and PolD polymerisation activities. Moreover, genetic analysis is used to address the in vivo relevance of the identified binding domain.

Because a complex interplay between the different polymerases in the replisome is key, particularly during lagging strand synthesis, it is important to understand this RPA-polymerase communication in order to understand how the replisome works. Because RPA is a main replisome protein to control polymerases understanding its mechanistic role is an important aspect.

The main statements made in the manuscript are:

- RPA2-WH domain is essential for survival of archaeal cells.
- PolD and ProSL both bind to the WH domain of RPA2
- A previously undescribed WH domain interaction mode is involved in the interaction with PolD
- The RPA2-WH domain is required to stimulate PreSL primer extension activity, but not for primer synthesis. In contrast, RPA inhibits PolD extension activity (WH domain independently).
- A model is presented how PriSL may conduct synthesis using RPA-coated ssDNA templates.

From the perspective of a non-expert in structural biology, the experimental results seem of high quality. The data and the observed effects in biochemical experiments are largely convincing. In some places, I find that the conclusions drawn require a stronger experimental base. Aspects of the model appear speculative to me. This is detailed below.

The value of study lies in its in-detail characterisation of known concepts rather than in its uncovering of completely new concepts. Such in-detail studies are an important part of scientific progress.

Issues to be addressed:

- It is claimed that the RPA2-WH domain is essential in archaeal cells. The authors describe that knock-outs lacking the WHD of RPA2 could be generated. They conclude that the domain is essential for cell survival. Whether loss-of-function impacts replication is not addressed. The evidence presented is too weak and based on negative evidence only.

A conditional loss-of-function system, if available, would help clarify this issue.

The two described molecular activities, binding of PolD and PriSL, were not investigated as regards their relevance for survival/replication. The binding surfaces are distinct and should be mutated individually if this is technically feasible. The resulting phenotype(s) may be less severe and allow analysis of PolD and PriSL binding individually. Moreover, such

separate binding site analysis would substantiate the finding of PolD interaction with the WHD, for which no biochemical or biological relevance is presented in the manuscript.

- The authors claim that the RPA2-WH domain is required to stimulate PreSL extension activity. The activating activity of RPA on PriSL was known. New is the role of the WHD of RPA2. PolD extension activity is inhibited by RPA, which was also known before.

The structure of the WH domain bound to PriSL should be analysed as to the mechanism of polymerisation activation. Does the structure indicate an effect of WH domain binding on the polymerase active site or of allosteric mechanism? Discuss alternative activation mechanisms.

RPA stimulation of DNA polymerases has been described and models for the mechanism suggested. The authors should discuss these models and compare them with PriSL stimulation by RPA.

The priming activity of PriSL does not seem to be affected as suggested by an M13 ssDNA priming experiment. The effect on double stranded nucleic acid synthesis shown seems clear. However, interpretation is complicated by the fact that it is not distinguished between RNA primer synthesis and DNA synthesis. Specific RNA and DNA detection (for example using radioactive nucleotides) should make conclusions much clearer and should be done.

- A model for how PriSL extension displaces RPA from ssDNA during extension is presented. According to the model, PriSL translocates on the template ssDNA. Contact with the RPA2-WH domain weakens the RPA-ssDNA interaction, leading to RPA dissociation. That cycle starts again when PriSL meets the next RPA. Albeit intuitive, key points remain speculative. Is there any evidence for a weakening of RPA-ssDNA interaction by the WH domain? RPA-ssDNA binding experiments upon titrating in PriSL seem feasible. Comparison of RPA2-WT with RPA2 mutant without WH domain should address the involvement of the WH domain.

Does the activation of the polymerase activity of PriSL by WHD interaction have something to do with RPA displacement or are these completely separate characteristics?

It is then suggested that PriSL hands over to PolD using a mechanism that involves the WH domain of RPA2. Intuitive is a competition model. Indeed the authors mention a possible competition between the two polymerases for WHD binding as based on steric clashes between them. Competitive binding should be tested.

The authors please elaborate on what the potential biological meaning of PriSL activation vs PolD inhibition by RPA could be. Why does PreSL need activation of its extension activity whereas it is advantageous that RPA inhibits PolD?

Specific points:

- New binding mode of WHD (with PolD). Please make a statement about conservation of this site in a) other archaeal RPA2-WHDs, b) with eukaryotic RPA.

- The authors please be more clear in the introduction and discussion when using terms like 'conserved', 'homologous' etc. Please state what you are referring to, with bacteria, among archaea, with eukaryotes?

- Describe the primer extension and priming assays better. Which components are added? State that PCNA is not present. Analysis by denaturing gel electrophoresis to see only newly synthesised strands (extension) or native electrophoresis (priming).

- The recombinant WH domain is added in the biochemical experiments to mutant RPA2 lacking the WH domain. Adding excess recombinant WHD could be informative. This situation could reveal effects of the domain that are lost due to the missing linkage to the RPA2 N-terminal portion.

- page 9, lines 358-366: Please make the link with the current study clearer.

- line 374: '...this model supports...' unclear to me how it supports that. Please specify.

- line 378: '...inducing a weaker DNA binding mode...'. Is this speculative or is there any evidence?

Reviewer #2

(Remarks to the Author)

This article by the Sauguet team describes the structural basis of the archaeal replisome, focussing on polD interactions with RPA. This integrative and multidisciplinary study is of the highest technical calibre and reports novel findings of interest to

researchers with a focus on the molecular mechanisms of DNA synthesis, and the evolution of replication in the domains of life. The manuscript makes an excellent overall impression, with very nice illustrations supporting the points made by the authors. Something bad happened to panel k and l on page 28 of the combined pdf file. Even though this is not my field, I found the article made very interesting reading and I could spot no errors, technical or otherwise, which just leaves me to congratulate the authors on a beautiful piece of work!

Reviewer #3

(Remarks to the Author)

The manuscript titled "Communication between DNA polymerases and Replication Protein A within the archaeal replisome" by Markel et al reveals the roles of WH domain within Rpa2, which can interact with the two key actors of the replisome: the DNA primase (PriSL) and the replicative DNA polymerase (PolD). Combined with the biochemical assays and structural analysis, this study sheds light on the assembly and regulation of replisome in archaea and provides important clue for investigating the eukaryotic replication process. Overall, the approach is of high quality and it potentially deserves the publication in Nature Communications. However, a number of concerns need to be addressed before the publication :

Major Comments:

1. What is the exact role for the interaction between the WH domain of Rpa2 and PolD? It's interesting that the PolD interacts with the WH domain of Rpa2 via a novel binding surface different from the PriSL and the authors proposed that the RPA could potentially play a role in a polymerase-switch event between PriSL and PolD. While the position of WH domain of Rpa2 in class 2 exposes the PriSL-binding interface, did the authors try to get the complex of Rpa2WH (or RPA), PriSL, and PolD? It will provide crucial insight into the coordination among the RPA, PriSL, and PolD.
2. What's the exact product of primer extension activity assay with PriSL? Is it RNA? Why did the assay system include both dNTPs and rNTPs?
3. How does the RPA stimulate the primer extension activity of PriSL? The authors should explain why the RPA-deltaWH could inhibit the extension activity of PriSL despite it didn't direct interact with the PriSL and why the RPA-deltaWH+Rpa2WH couldn't rescue the stimulation effect of PriSL.

Minor comments :

1. Fig.2e is ahead of Fig.2d.
2. It is strange that the label of Primase in Figures is not consistent with the result section (PriSL).
3. Line135-136: "RPA-bound nucleoprotein filaments lacking Rpa2WH displayed 7-fold and 4-fold weaker interactions with PriSL and PolD respectively (Fig.3b,3d)"—The single point could not support the fitness of the binding affinity in BLI assay.
4. Fig.4: The quantification of the inhibition efficiency by RPA is strange. It's not gradually decreased by the increased concentration of RPA or Rpa2deltaWH. The Rpa2WH seems exhibit a weaker inhibitory effect.
5. Fig.5-6: The electrostatic interactions should be shown by the lines to make it clearer.
6. The Supplementary figures should contain the representative densities of the solved structures, especially the interaction interfaces.

Version 1:

Reviewer comments:

Reviewer #1

(Remarks to the Author)

The revised manuscript is clearer and more precisely written. It will be fit for publication if a few issues left are addressed.

Model figure 8b and discussion:

The authors discuss at length a potential handover-model of RPA-WH from PriSL to PolD. Although interesting in light of the data presented, this suffers from unclarity about the physiological importance of the PolD-RPA-WH interaction. The authors show no effect of abrogating the interaction on the RPA effect on PolD polymerisation. Please make clear what we know about the physiological importance of the interaction.

Presentation of on vitro polymerisation experiments:

It is important that also non-insiders can understand how the experiments were done. Please go through Fig 4 again and add important experimental detail to the main text, figure and figure legend.

- At the beginning of the section, please name all important reaction components added and other important experimental parameters such as reaction time.
 - Describe the bands that can be seen in the gel images. For example, 4a-d look quite different to e-h although the full-length product should be the same 87 nt band. Also, in e-h is a pronounced double band, of which the lower one gets fainter with increasing RPA (e) or both get fainter (g,h). Explain please.
- In addition, please improve the description of the bar graphs. Which bands were quantified for them? The upper band? The lower band? Both? State what the error bars are. How many times were the experiments made?
- The cartoons going with the panels are useful. Please provide a legend for the symbols. Ideally add the DNA configuration as a cartoon too.

Fig 2:

Adding the amino acid positions of the domains in the domain models would help.

Reviewer #3

(Remarks to the Author)

The authors have addressed all my concerns and I strongly recommended it for publication in Nature Communications.

We would like to take the opportunity to sincerely thank the reviewers for their enthusiastic comments about our work, and for the important points that they raised in order to help us improve the manuscript. We believe that the manuscript has been improved considerably thanks to their feedbacks.

Point-by-point answers to the reviewer's comments (answers colored in blue):

Reviewer #1:

The authors describe structurally how archaeal RPA2 binds PriSL, archaeal primase, and PolD, the replicative polymerase, as a means to better understand the communication between polymerases and RPA in the replisome, as stated in the title. The WH domain of RPA2 is identified as the main interaction site. The authors then use in vitro analysis to characterise the effect of the critical binding determinants in the RPA2-WH domain on PriSL and PolD polymerisation activities. Moreover, genetic analysis is used to address the in vivo relevance of the identified binding domain.

Because a complex interplay between the different polymerases in the replisome is key, particularly during lagging strand synthesis, it is important to understand this RPA-polymerase communication in order to understand how the replisome works. Because RPA is a main replisome protein to control polymerases understanding its mechanistic role is an important aspect.

The main statements made in the manuscript are:

- RPA2-WH domain is essential for survival of archaeal cells.
- PolD and PriSL both bind to the WH domain of RPA2
- A previously undescribed WH domain interaction mode is involved in the interaction with PolD
- The RPA2-WH domain is required to stimulate PriSL primer extension activity, but not for primer synthesis. In contrast, RPA inhibits PolD extension activity (WH domain independently).
- A model is presented how PriSL may conduct synthesis using RPA-coated ssDNA templates.

From the perspective of a non-expert in structural biology, the experimental results seem of high quality. The data and the observed effects in biochemical experiments are largely convincing. In some places, I find that the conclusions drawn require a stronger experimental base. Aspects of the model appear speculative to me. This is detailed below.

The value of study lies in its in-detail characterisation of known concepts rather than in its uncovering of completely new concepts. Such in-detail studies are an important part of scientific progress.

Issues to be addressed:

- It is claimed that the RPA2-WH domain is essential in archaeal cells. The authors describe that knock-outs lacking the WHD of RPA2 could be generated. They conclude that the domain is essential for cell survival. Whether loss-of-function impacts replication is not addressed. The evidence presented is too weak and based on negative

evidence only.

A conditional loss-of-function system, if available, would help clarify this issue.

Maybe there was a misunderstanding or a typo in the reviewer's comment, but we describe that knock-outs lacking the WH domain of Rpa2 could **not** be generated. It is important to note that all previous genetic studies on RPA in Archaea focused on the deletion of entire subunits. This study is the first to investigate the role of individual RPA domains in Archaea using genetic approaches.

Thermococcus barophilus has been developed in the lab as a genetic model to study Thermococcales archaea but no conditional loss-of function system is available yet on this model. Nevertheless, using *Thermococcus barophilus* as a genetic model, we have managed to knock out proteins involved in DNA replication and repair like family B DNA polymerase and Ribonuclease HII, while deletion of family D DNA polymerase never succeeded (DOI: [10.3390/genes9020077](https://doi.org/10.3390/genes9020077)). In this methodology, screening of 5-10 clones was sufficient to select viable deletion mutants. In the current study, 37 clones were screened for the deletion of the Rpa2 WH domain. Using a similar methodology, the possible essentiality of PoID in *T. kodakarensis* (doi: [10.1128/JB.02037-12](https://doi.org/10.1128/JB.02037-12)) was reported by another group.

In the revised manuscript, we clarify that the Rpa2WH domain is "potentially" essential. We have edited the text: Page 3, line 120; Page 4, line 138 and Page 9, line 419.

Additionally, we cite a genome-scale analysis of gene function in the methanogenic archaeon *M. maripaludis* from the Whitman lab, which further supports our findings (<https://doi.org/10.1073/pnas.1220225110>).

Page 4, lines 132-134 : “*A genetic study on the methanogenic archaeon Methanococcus maripaludis found that rpa1 and rpa2 are possibly essential genes, whereas rpa3 is not, highlighting the critical biological role of Rpa2³³.*”

The two described molecular activities, binding of PoID and PriSL, were not investigated as regards their relevance for survival/replication. The binding surfaces are distinct and should be mutated individually if this is technically feasible. The resulting phenotype(s) may be less severe and allow analysis of PoID and PriSL binding individually. Moreover, such separate binding site analysis would substantiate the finding of PoID interaction with the WHD, for which no biochemical or biological relevance is presented in the manuscript.

We agree with reviewer #1 in that the suggested experiments would significantly expand on our findings. In practice, this would require designing mutations that would specifically alter the interactions between the WH domain and each polymerase without altering the integrity nor stability of the WH domain itself, as a less stable WH variant would likely affect both interactions. Therefore, a specific set of mutants for PriSL-WH and PoID-WH complexes should be iteratively designed and produced to validate *in vitro* that they only affect the interaction with their corresponding polymerase. After that, these mutations could be considered *in vivo*. However, such mutations may also be lethal.

Altogether we agree with reviewer #1 in that the proposed experiments could eventually shed light on the role of the interaction between Rpa2 and PoID *in vivo*, but we

believe they are beyond the scope of this study where the main message is the identification of the interactions between RPA, PriSL and PolD. We have modified the manuscript mentioning potential interesting perspectives for future investigation of the role of RPA binding to PolD (p. 12, lines 617-619):

Further insights into the RPA-PolD interaction will be needed to understand its role in the replisome, with non-lethal mutants of the Rpa2-PolD interface or with single-molecule studies to track potential polymerase switch events in presence of RPA.'

- The authors claim that the RPA2-WH domain is required to stimulate PriSL extension activity.

The activating activity of RPA on PriSL was known. New is the role of the WHD of RPA2. PolD extension activity is inhibited by RPA, which was also known before.

The structure of the WH domain bound to PriSL should be analysed as to the mechanism of polymerisation activation. Does the structure indicate an effect of WH domain binding on the polymerase active site or of allosteric mechanism? Discuss alternative activation mechanisms.

In this study we report the first crystal structures of the *Pyrococcus abyssi* primase (PriSL)PriSL both in complex with the Rpa2 WH domain and in its apo form. **Supplementary Figure 9 (see below)** has been extensively reshaped and extended to discuss the comparison between our crystal structures of PriSL and in complex with the Rpa2^{WH} domain. The catalytic PriS subunit does not undergo any conformational changes in the presence of the Rpa2^{WH} domain. Furthermore, the primer-extension activity assays do not show any observable effect of the isolated WH domain on primase activity (neither stimulation nor inhibition). Taken together, these experiments rule out an allosteric activation mechanism. We have extended the discussion section to clarify this conclusion, and to carefully revise and phrase our conclusions from the primer extension assay with several RPA mutants (p. 10, lines 464-471):

'We also considered the possibility of an allosteric stimulatory effect of Rpa2^{WH} on PriSL. Comparing the crystal structures of PriSL and the PriSL-Rpa2^{WH} complex, the active site in PriS does not undergo significant conformational changes upon binding of Rpa2^{WH} (Supplementary Fig. 9d). Taken together with the finding that Rpa2^{WH} alone does not stimulate primer elongation by PriSL (Fig. 4f), we conclude that there is no allosteric activation of PriSL by Rpa2^{WH}. As truncated RPA lacking the WH domain did not stimulate PriSL, our results indicate that Rpa2^{WH} is required but not sufficient for primer elongation stimulation. Rpa2^{WH} must also be tethered to the RPA trimeric core for the PriSL stimulatory effect to take place.'

Supplementary Figure 9: Structural conservation of the heterodimeric euryarchaeal DNA primase. (a) Model of *Pyrococcus abyssi* PriS-PriL^{ΔCTD} crystal structure at 1.85 Å resolution (PabPriSL^{ΔCTD}). (b) Alignment of human primase (PDB ID 4BPW) to PabPriSL^{ΔCTD}. (c) Alignment of *Saccharolobus solfataricus* PriSLX (PDB ID 5OFN) to PabPriSL^{ΔCTD}. (d) Superposition of the crystal structures of PabPriSL^{ΔCTD} (orange) and the Rpa2^{WH}-PriSL^{ΔCTD} complex (green), illustrating that PriS does not undergo significant conformational changes upon binding to Rpa2.

RPA stimulation of DNA polymerases has been described and models for the mechanism suggested. The authors should discuss these models and compare them with PriSL stimulation by RPA.

We have included a comparison of the archaeal RPA-PriSL interaction mechanism with the interaction mechanism of human RPA with PrimPol (p. 11, lines 564-572), as well as with the interaction of yeast RPA with Pol- α /primase (p. 9 lines 438-443). After the submission of our manuscript, we found a preprint on biorxiv with a study on the effects of human RPA on the activity of the primosome. The discussion has been extended to compare our results with those of that study (p. 11, lines 575-581):

'During the submission of this article, a pre-print was published with interaction studies between human RPA and the primosome (the Pol- α /primase complex). In that study, human RPA reportedly stimulates primosome activity on DNA hairpins caused by inverted sequence repeats. Additionally, this stimulatory effect is lost in RPA mutants lacking the conserved WH domain. It is noteworthy that they found that human RPA has an inhibitory effect on primosome activity on unstructured DNA. Nevertheless, and in agreement with our findings on archaeal RPA, this inhibitory effect was increased in RPA mutants lacking the WH domain.'

The priming activity of PriSL does not seem to be affected as suggested by an M13 ssDNA priming experiment. The effect on double stranded nucleic acid synthesis shown seems clear. However, interpretation is complicated by the fact that it is not distinguished between RNA primer synthesis and DNA synthesis. Specific RNA and DNA detection (for example using radioactive nucleotides) should make conclusions much clearer and should be done.

While the eukaryotic PriSL acts as a specific RNA polymerase, synthesizing a short RNA primer, the archaeal primase has been shown to synthesize a hybrid RNA/DNA mixed primer (Greci et al, Nat. communications 2022 DOI: s41467-022-28093-2). Therefore, our *in vitro* activity assays were conducted using a mix of dNTPs and rNTPs that mimics the nucleotide concentrations found in *Pyrococcus abyssi* *in vivo*. The concentrations of dNTPs and rNTPs used in this study were experimentally determined in one of our previous studies (Lemor et al, J. Mol. Biol (2018) DOI: [10.1016/j.jmb.2018.10.004](https://doi.org/10.1016/j.jmb.2018.10.004)).

Nevertheless, we agree with reviewers #1 (and also reviewer #3) in that the distinction between RNA and DNA synthesis is crucial when discussing the archaeal PriSL. To further investigate this point, we have performed additional experiments and revised the manuscript accordingly with a new results section (page 6, lines 227-250).

Regarding the primer-extension activity assays: To discriminate between RNA and DNA products synthesized by PriSL, we treated primer extension reactions with 250 mM NaOH at 55°C for 2 h. As demonstrated in a control experiment with dsDNA including a single riboadenosine nucleotide (**Supplementary Figure 7i**), this treatment cleaves oligonucleotides containing ribonucleotides. Negative controls treated with 250 mM NaCl instead remain uncleaved. All primer-extension experiments in presence of RPA variants previously presented in **Figure 4** have been reproduced. Resulting reactions were incubated at 55°C for 2 hours with either 250mM NaCl or 250mM NaOH (**Supplementary Figure 7 a-h**). As all reaction products remained uncleaved by alkaline treatment, these experiments confirm that PabPriSL extends

a primer using dNTPs in the presence or in the absence of wild type RPA or any of the tested variants. This result is consistent with former studies from our group (Le Breton et al, JMB 2007: DOI: [10.1016/j.jmb.2007.10.015](https://doi.org/10.1016/j.jmb.2007.10.015) & Lemor et al, JMB 2018: DOI: [10.1016/j.jmb.2018.10.004](https://doi.org/10.1016/j.jmb.2018.10.004)) and others have shown that the primase from euryarchaea primarily extends a primer using dNTPs (Liu et al, JBC 2001, DOI: <https://doi.org/10.1074/jbc.M10639120>).

Supplementary Figure 7. Preference for the incorporation of deoxyribonucleotides versus ribonucleotides to primers by PriSL during elongation. Reactions identical to primer extension assays in presence of RPA variants (Figure 4) are subjected to a treatment with (a, c, e, g) 250 mM NaCl or (b, d, f, h) 250 mM NaOH and incubated at 55°C for 2 hours. (i) Treatment with NaOH cleaves oligonucleotides containing ribonucleotides, as demonstrated with a FAM-labeled 34-nucleotide dsDNA substrate containing a riboadenosine in the 8th position (Supplementary Table 4).

Regarding the priming experiments: Similar to the primer-extension activity assay, our original priming activity assay was performed using a mix of dNTPs and rNTPs that mimics the nucleotide concentrations found in *Pyrococcus abyssi in vivo* (Lemor et al., J. Mol. Biol., 2018, DOI: 10.1016/j.jmb.2018.10.004). We have now repeated the assays, this time exclusively in the presence of rNTPs or dNTPs. The results of these assays have been compiled in a new **Supplementary Figure 6** (see below). In both cases, the synthesis of longer primers by the primase is enhanced by the addition of RPA, although RNA primer synthesis is weaker due to the euryarchaeal primase's preference for dNTPs over rNTPs (as discussed earlier).

In conclusion, these additional experiments demonstrate that PriSL preferentially elongates primers with dNTPs, and that RPA stimulates primer extension by PriSL regardless of the nucleotide combination used: dNTPs, rNTPs, or a mix of dNTPs+rNTPs.

Supplementary Figure 6: Impact of RPA binding on PriSL priming activity. M13mp18 circular ssDNA was incubated with PriSL and increasing amounts of RPA (0-6.4 μM) in the presence of dNTPs+rNTPs (a), dNTPs only (b) or rNTPs only (c). Control lanes contain oligonucleotide 1 kb Plus DNA Ladder or Low Range ssRNA Ladder. For details, see the method section.

- A model for how PriSL extension displaces RPA from ssDNA during extension is presented. According to the model, PriSL translocates on the template ssDNA. Contact with the RPA2-WH domain weakens the RPA-ssDNA interaction, leading to RPA dissociation. That cycle starts again when PriSL meets the next RPA.

Albeit intuitive, key points remain speculative. Is there any evidence for a weakening of RPA-ssDNA interaction by the WH domain? RPA-ssDNA binding experiments upon titrating in PriSL seem feasible. Comparison of RPA2-WT with RPA2 mutant without WH domain should address the involvement of the WH domain.

Under the conditions of our BLI experiments, the binding of PriSL to RPA-ssDNA filaments does not cause any observable weakening of the RPA-ssDNA interaction. Indeed, RPA-ssDNA filaments remain stable in the presence of saturating concentrations of PriSL and during a long period of association time (30 minutes, **Figure 3b**). This is consistent with our proposed mechanism: we hypothesize that only while the primer is extended by PriSL, the interaction of PriSL with the WH domain will weaken the RPA-ssDNA interaction of the downstream RPA only.

While directly measuring the displacement of RPA by PriSL is technically challenging, we believe that our primer-extension assays with multiple RPA variants demonstrate that PriSL can interact with the WH domain of RPA in order to promote RPA clearance from ssDNA. This kind of WH-mediated hand-off of RPA-ssDNA has been reported by others in the case of Rad51-RPA interactions (reference 56), as well as in Rad52-RPA mediated complementary ssDNA annealing (reference 57). In the future, performing single-molecule studies with RPA-coated ssDNA and DNA polymerases hold great potential to further understand these interactions.

Does the activation of the polymerase activity of PriSL by WHD interaction have something to do with RPA displacement or are these completely separate characteristics?

As PriSL elongates the primer, it interacts with the downstream WH promoting the clearance of the downstream RPA. In our primer extension assays, RPA lacking the WH domain cannot be displaced, resulting in early termination of primer elongation by PriSL. This indicates that the WH domain of Rpa2 synchronizes the primer elongation by PriSL with the displacement of RPA.

Nevertheless, there is an additional component to the WH-mediated PriSL stimulation besides the RPA clearance, as full-length RPA stimulates PriSL primer extension compared to the reactions in the absence of RPA. As we measured the affinity of the interaction between Rpa2^{WH} and PriSL to be 24 ± 2 nM, this stimulatory effect could be achieved by enriching the local concentration of PriSL around RPA-coated ssDNA.

It is then suggested that PriSL hands over to PolD using a mechanism that involves the WH domain of RPA2. Intuitive is a competition model. Indeed the authors mention a possible competition between the two polymerases for WHD binding as based on steric clashes between them. Competitive binding should be tested.

The study of a potential competitive binding model between Rpa2^{WH}, PriSL and PolD is complicated by the fact that PriSL and PolD bind together with high affinity (Madru et al,

Nature communications 2020: <https://www.nature.com/articles/s41467-020-15392-9>; Oki et al, NAR 2021: <https://doi.org/10.1093/nar/gkab243>).

It is noteworthy that the PriSL-Rpa2^{WH} complex, when superposed with the PolD-Rpa2^{WH} complex, would have to transfer the dsDNA to the active site of PolD over 90 Å away (**Fig. 7a**). This suggests that the entire hand-off process is likely to also be driven by additional molecular interactions, either directly between PriSL and PolD or with the contribution of additional replication or repair factors. The manuscript has been updated to reflect this point (page 12 lines 629-635):

'It is noteworthy that the PriSL-Rpa2^{WH} complex, when superposed with the PolD-Rpa2^{WH} complex, would have to transfer the dsDNA to the active site of PolD over 90 Å away (Fig. 7a). This suggests that the entire hand-off process is likely to also be driven by additional molecular interactions, either directly between PriSL and PolD or with the contribution of additional replication or repair factors. Investigating this transient potentially RPA-mediated polymerase switch will be crucial to understand the dynamic nature of the archaeal replisome.'

The authors please elaborate on what the potential biological meaning of PriSL activation vs PolD inhibition by RPA could be. Why does PriSL need activation of its extension activity whereas it is advantageous that RPA inhibits PolD?

As the main function of PriSL is to prime (or re-prime following replication stress) ssDNA, it seems appropriate that RPA-bound ssDNA stretches would recruit PriSL and stimulate its activity *in vivo*. On the other hand, PolD can only extend dsDNA, which is handed-off by PriSL during lagging-strand replication. In the original manuscript, we mentioned the fact that during DNA replication PolD might not be inhibited by RPA and is likely to exhibit a more complex behavior *in vivo*, as it acts in complex with PCNA (p11, lines 593-596).

Specific points:

- New binding mode of WHD (with PolD). Please make a statement about conservation of this site in a) other archaeal RPA2-WHDs, b) with eukaryotic RPA.

We have made a new supplementary figure with a structural MSA made with FoldMason (DOI:[10.1101/2024.08.01.606130](https://doi.org/10.1101/2024.08.01.606130)) including Rpa2 WH domains from several Asgard Archaea, thermococcales, yeast and human sequences (**Supplementary Figure 12**, see below). We have highlighted the residues that contact PriSL and PolD respectively, and indicated their position in the secondary structure of the WH domains. This combined structural and sequence-conservation analysis reveals the following:

- The structure of the WH domain at the interface with the primase is highly conserved. Although sequence conservation with the eukaryotic WH domain is weaker than among Archaea, Y258 is almost fully conserved in our alignment. We have previously shown that the PriSL-interacting interface on Rpa2 WH is conserved in eukaryotes, despite the lack of strong sequence conservation (**Fig. 6**).
- The structure and interacting residues of the WH domain at the PolD interface are well conserved within Thermococcales, but not with eukaryotes. This is not unexpected, since PolD does not exist in eukaryotes.

Supplementary Figure 12. Structural alignment of Rpa2 WH domains from archaea and eukaryotic sources. Experimental structures were used when available, and AlphaFold predictions were used for the rest of queries. Sequence conservation is indicated with grey shading. Positions in *P. abyssi* Rpa2 that contact PolD and PriSL are indicated with blue and orange circles respectively.

- The authors please be more clear in the introduction and discussion when using terms like 'conserved', 'homologous' etc. Please state what you are referring to, with bacteria, among archaea, with eukaryotes?

We have revised carefully the manuscript to distinguish which features are conserved within the different domains of life.

- Describe the primer extension and priming assays better. Which components are added?

State that PCNA is not present. Analysis by denaturing gel electrophoresis to see only newly synthesized strands (extension) or native electrophoresis (priming).

We have revised the corresponding results section of the manuscript accordingly, elaborating on the experiment setup and explaining the accumulation of short products in reactions with PriSL and short degradation products in reactions with PolD. As mentioned above, we have also performed additional experiments to document the effect of RPA on RNA or DNA primer synthesis.

- The recombinant WH domain is added in the biochemical experiments to mutant RPA2 lacking the WH domain. Adding excess recombinant WHD could be informative. This situation could reveal effects of the domain that are lost due to the missing linkage to the RPA2 N-terminal portion.

In order to further verify the role of the Rpa2^{WH}-PriSL interaction, we repeated the primer extension assay with wild-type RPA and PriSL, in the presence of 1, 2.5, 5, 10 and 20-fold molar excess of Rpa2^{WH} as suggested by reviewer #1. We expected the results to be similar to the experiment with truncated RPA- Δ WH (**Figure 4 g, h**), however the stimulation of PriSL primer extension was only slightly lower than in the control reactions. The most noticeable effect was in the very longest extension products, which gradually decreased with higher excess of Rpa2^{WH} (see black arrowhead in gel below).

In reactions with RPA- Δ WH or RPA- Δ WH + Rpa2^{WH}, PriSL stimulation was severely reduced in a concentration-dependent manner (**Fig. 4 g,h**), indicating the importance of the PriSL-WH interaction. Our interpretation of the result with wtRPA + excess Rpa2^{WH} is that even in the presence of free Rpa2^{WH}, as PriSL elongates the primer the local concentration of downstream RPA is sufficient for PriSL to eventually bind to the WH domain of RPA and benefit from its stimulatory effect. The DNA substrate will locally enrich both PriSL and wild-type RPA. Nevertheless, a small effect is noticeable in the absolutely longest extension products.

- page 9, lines 358-366: Please make the link with the current study clearer.

The paragraph has been rephrased to clarify the link between the cited studies and our manuscript (p. 9-10 lines 438-455):

'In the context of DNA replication, eukaryotic RPA has been shown to interact with Pol- α /primase in the lagging strand. Deletion of the Rfa2 WH domain, orthologous to archaeal Rpa2^{WH}, results in reduced interaction between RPA and Pol- α /primase⁴⁹. More recently, the Rfa1 OB-F domain has been shown to negatively affect lagging-strand replication in vitro, while the Rfa2 WH domain positively affected it, suggesting that the two domains act in concert to regulate priming frequency in the replisome. Our results indicate that the archaeal Rpa2 WH domain interacts with PriSL as well, whose primer extension activity was stimulated by RPA in a concentration-dependent manner (Fig. 4e-h).'

- line 374: '...this model supports...' unclear to me how it supports that. Please specify.

This explanation has been rephrased to better reflect the contribution of the structural model to our hypothesis (p. 10, lines 488-492):

'This model shows the relative orientation of RPA and PriSL when they interact during primer extension, indicating that RPA could optimally tether PriSL to the downstream ssDNA via the WH domain. Based on this model and our primer extension results, we have proposed a PriSL-RPA 'WH-bait' interaction model mechanism (Fig. 8b), explaining how RPA can stimulate PriSL.'

- line 378: '...inducing a weaker DNA binding mode...'. Is this speculative or is there any evidence?

There is evidence of a similar mode of action in the interaction between Rad51 and RPA. The text has been updated to reflect this (page 10, lines 494-465):

'...possibly inducing a weaker DNA-binding mode on the bound RPA as has been observed in the Rad51-RPA interaction⁵⁶, and ultimately displacing it RPA from the template strand.'

Reviewer #2 (Remarks to the Author):

This article by the Sauget team describes the structural basis of the archaeal replisome, focussing on polD interactions with RPA. This integrative and multidisciplinary study is of the highest technical calibre and reports novel findings of interest to researchers with a focus on the molecular mechanisms of DNA synthesis, and the evolution of replication in the domains of life. The manuscript makes an excellent overall impression, with very nice illustrations supporting the points made by the authors. Something bad happened to panel k and l on page 28 of the combined pdf file. Even though this is not my field, I found the article made very interesting reading and I could spot no errors, technical or otherwise, which just leaves me to congratulate the authors on a beautiful piece of work!

We greatly appreciate the enthusiastic feedback from reviewer #2. We have fixed Figure 3 so that panels k and l are properly visible in the revised manuscript.

Reviewer #3 (Remarks to the Author):

The manuscript titled "Communication between DNA polymerases and Replication Protein A within the archaeal replisome" by Markel et al reveals the roles of WH domain within Rpa2, which can interact with the two key actors of the replisome: the DNA primase (PriSL) and the replicative DNA polymerase (PolD). Combined with the biochemical assays and structural analysis, this study sheds light on the assembly and regulation of replisome in archaea and provides important clue for investigating the eukaryotic replication process. Overall, the approach is of high quality and it potentially deserves the publication in Nature Communications. However, a number of concerns need to be addressed before the publication :

Major Comments:

1. What is the exact role for the interaction between the WH domain of Rpa2 and PolD? It's interesting that the PolD interacts with the WH domain of Rpa2 via a novel binding surface different from the PriSL and the authors proposed that the RPA could potentially play a role in a polymerase-switch event between PriSL and PolD. While the position of WH domain of Rpa2 in class 2 exposes the PriSL-binding interface, did the authors try to get the complex of Rpa2WH (or RPA), PriSL, and PolD? It will provide crucial insight into the coordination among the RPA, PriSL, and PolD.

We have extensively tried to reconstitute a PriSL-RPA-PolD complex for structural studies without success. Our structural analysis of the PriSL-Rpa2^{WH} and the PolD-Rpa2^{WH} complexes shows that the polymerases would clash in such a ternary complex, which leads us to speculate that if the WH domain facilitates a polymerase switch event, it only engages both polymerases transiently. We agree with reviewer #3 in that investigating this mechanism further and with other methodologies (such as single-molecule fluorescence studies) will provide very interesting insights into RPA-mediated polymerase coordination, and have updated the text to reflect this (p12, lines 629-635).

2. What's the exact product of primer extension activity assay with PriSL? Is it RNA? Why did the assay system include both dNTPs and rNTPs?

While the eukaryotic PriSL acts as a specific RNA polymerase, synthesizing a short RNA primer, the archaeal primase has been shown to synthesize a hybrid RNA/DNA mixed primer (Greci et al, Nat. communications 2022 DOI: s41467-022-28093-2). Therefore, our *in vitro* activity assays were conducted using a mix of dNTPs and rNTPs that mimics the nucleotide concentrations found in *Pyrococcus abyssi in vivo*. The concentrations of dNTPs and rNTPs used in this study were experimentally determined in one of our previous studies (Lemor et al, J. Mol. Biol (2018) DOI: [10.1016/j.jmb.2018.10.004](https://doi.org/10.1016/j.jmb.2018.10.004)).

Nevertheless, we agree with reviewers #3 (and also reviewer #1) in that the distinction between RNA and DNA synthesis is crucial when discussing the archaeal PriSL. To further investigate this point, we have performed additional experiments and revised the manuscript accordingly with a new results section (page 6, lines 227-250):

Regarding the primer-extension activity assays: To discriminate between RNA and DNA products synthesized by PriSL, we treated primer extension reactions with 250 mM NaOH at 55°C for 2 h. As demonstrated in a control experiment with dsDNA including a single riboadenosine nucleotide (**Supplementary Figure 7i**), this treatment cleaves oligonucleotides containing ribonucleotides. Negative controls treated with 250 mM NaCl instead remain uncleaved. All primer-extension experiments in presence of RPA variants previously presented in **Figure 4** have been reproduced. Resulting reactions were incubated at 55°C for 2 hours with either 250mM NaCl or 250mM NaOH (**Supplementary Figure 7 a-h**). As all reaction products remained uncleaved by alkaline treatment, these experiments confirm that PabPriSL extends a primer using dNTPs in the presence or in the absence of wild type RPA or any of the tested variants. This result is consistent with former studies from our group (Le Breton et al, JMB 2007: DOI: [10.1016/j.jmb.2007.10.015](https://doi.org/10.1016/j.jmb.2007.10.015) & Lemor et al, JMB 2018: DOI: [10.1016/j.jmb.2018.10.004](https://doi.org/10.1016/j.jmb.2018.10.004)) and others have shown that the primase from euryarchaea primarily extends a primer using dNTPs (Liu et al, JBC 2001, DOI: <https://doi.org/10.1074/jbc.M10639120>).

Supplementary Figure 7. Preference for the incorporation of deoxyribonucleotides versus ribonucleotides to primers by PriSL during elongation. Reactions identical to primer extension assays in presence of RPA variants (Figure 4) are subjected to a treatment with (a, c, e, g) 250 mM NaCl or (b, d, f, h) 250 mM NaOH and incubated at 55°C for 2 hours. (i) Treatment with NaOH cleaves oligonucleotides containing ribonucleotides, as demonstrated with a FAM-labeled 34-nucleotide dsDNA substrate containing a riboadenosine in the 8th position (Supplementary Table 4).

Regarding the priming experiments: Similar to the primer-extension activity assay, our original priming activity assay was performed using a mix of dNTPs and rNTPs that mimics the nucleotide concentrations found in *Pyrococcus abyssi* *in vivo* (Lemor et al., J. Mol. Biol., 2018, DOI: 10.1016/j.jmb.2018.10.004). We have now repeated the assays, this time exclusively in the presence of rNTPs or dNTPs. The results of these assays have been compiled in a new **Supplementary Figure 6** (see below). In both cases, the synthesis of

longer primers by the primase is enhanced by the addition of RPA, although RNA primer synthesis is weaker due to the euryarchaeal primase's preference for dNTPs over rNTPs (as discussed earlier).

In conclusion, these additional experiments demonstrate that PriSL preferentially elongates primers with dNTPs, and that RPA stimulates primer extension by PriSL regardless of the nucleotide combination used: dNTPs, rNTPs, or a mix of dNTPs+rNTPs.

Supplementary Figure 6: Impact of RPA binding on PriSL priming activity. M13mp18 circular ssDNA was incubated with PriSL and increasing amounts of RPA (0-6.4 μM) in the presence of dNTPs+rNTPs (a), dNTPs only (b) or rNTPs only (c). Control lanes contain oligonucleotide 1 kb Plus DNA Ladder or Low Range ssRNA Ladder. For details, see the method section.

3. How does the RPA stimulate the primer extension activity of PriSL? The authors should explain why the RPA-deltaWH could inhibit the extension activity of PriSL despite it didn't direct interact with the PriSL and why the RPA-deltaWH+Rpa2WH couldn't rescue the stimulation effect of PriSL.

The WH-mediated stimulation of primer extension by PriSL appears to have two components, as evidenced by our primer extension assays:

- Truncated RPA lacking the WH domain results in early termination of primer extension by PriSL, indicating that it blocks the progression of PriSL. Therefore, the PriSL-WH interaction synchronizes RPA displacement with the primer elongation by PriSL.
- Full-length RPA stimulates primer extension by PriSL compared to reactions in the absence of RPA, indicating that it stimulates PriSL through a mechanism other than triggering its clearance from ssDNA upon binding to PriSL. Furthermore, since our crystal structures of PriSL and the PriSL-Rpa2^{WH} complex do not reveal any conformational changes in the catalytic PriS subunit upon binding of Rpa2^{WH}, we conclude that RPA recruits PriSL to ssDNA more efficiently than PriSL by itself can bind to ssDNA.

Minor comments :

1. Fig.2e is ahead of Fig.2d.

The typo was corrected and the figure was properly referenced in the text as Figure 2c.

2. It is strange that the label of Primase in Figures is not consistent with the result section (PriSL).

We have homogenized the label of Primase to PriSL in all figures.

3. Line135-136: “RPA-bound nucleoprotein filaments lacking Rpa2WH displayed 7-fold and 4-fold weaker interactions with PriSL and PolD respectively (Fig.3b,3d)” —The single point could not support the fitness of the binding affinity in BLI assay.

We agree with reviewer #3 in that the wording might have been misleading. To clarify that the shown experiment does not reflect a full steady-state analysis, this section has been rephrased as shown below (p. 4, lines 149-151):

‘Additionally, RPA-bound nucleoprotein filaments lacking Rpa2^{WH} displayed 7-fold and 4-fold weaker interactions with PriSL and PolD respectively under the tested conditions (12.5 nM PriSL in Fig. 3b, 250 nM PolD in Fig. 3d).’

4. Fig.4: The quantification of the inhibition efficiency by RPA is strange. It’s not gradually decreased by the increased concentration of RPA or Rpa2deltaWH. The Rpa2WH seems exhibit a weaker inhibitory effect.

We decided to quantify the primer elongation efficiency by PriSL by integrating the longest extension products, where the trend of RPA concentration-dependent stimulation is clear. For the sake of consistency, we quantified the effect of RPA on PolD in a similar manner. However, when PolD stalls due to obstruction by RPA, shorter digested products are accumulated due to its increased exonuclease activity. We believe that this effect leads to a non-gradual observed decrease of its elongation activity as a function of RPA concentration. The manuscript has been updated to include this point (page 5, lines 194-196).

5. Fig.5-6: The electrostatic interactions should be shown by the lines to make it clearer.

Figures 5 and 6 have been modified accordingly.

6. The Supplementary figures should contain the representative densities of the solved structures, especially the interaction interfaces.

We have added **Supplementary figure 8** (see below) showing additional representative density for all structures, especially in the regions of intermolecular interactions.

Supplementary Figure 8: Illustration of the quality of the X-ray crystallography electron density map and the cryo-EM maps, in the interfacial region with the Rpa2 winged-helix domain. Left panels show the 2mFo-Fc electron density map contoured at 1.0 σ for the PriS-PriL^{ACTD}-Rpa2^{WH} ternary complex. Middle (class1) and right (class2) panels show the cryo-EM maps contoured at a threshold of 0.148 for the DP1-DP2-Rpa2^{WH} ternary complexes.